# ParaRNN: Unlocking Parallel Training of Nonlinear RNNs for Large Language Models

**Federico Danieli**   **Pau Rodríguez**   **Miguel Sarabia**   **Xavier Suau**   **Luca Zappella**

Apple

`f_danieli@apple.com`

## Abstract

Recurrent Neural Networks (RNNs) laid the foundation for sequence modeling, but their intrinsic sequential nature restricts parallel computation, creating a fundamental barrier to scaling. This has led to the dominance of parallelizable architectures like Transformers and, more recently, State Space Models (SSMs). While SSMs achieve efficient parallelization through structured linear recurrences, this linearity constraint limits their expressive power and precludes modeling complex, nonlinear sequence-wise dependencies. To address this, we present ParaRNN, a framework that breaks the sequence-parallelization barrier for nonlinear RNNs. Building on prior work, we cast the sequence of nonlinear recurrence relationships as a single system of equations, which we solve in parallel using Newton's iterations combined with custom parallel reductions. Our implementation achieves speedups of up to $665\times$ over naïve sequential application, allowing training nonlinear RNNs at unprecedented scales. To showcase this, we apply ParaRNN to adaptations of LSTM and GRU architectures, successfully training models of 7B parameters that attain perplexity comparable to similarly-sized Transformers and Mamba2 architectures. To accelerate research in efficient sequence modeling, we release the ParaRNN codebase as an open-source framework for automatic training-parallelization of nonlinear RNNs, enabling researchers and practitioners to explore new nonlinear RNN models at scale.

## 1 Introduction

Since its introduction by Vaswani et al. (2017), the Transformer architecture has quickly imposed itself as the *de-facto* choice for sequence modeling, surpassing previous state-of-the-art, RNN-based models such as GRU and LSTM (Cho et al., 2014; Hochreiter & Schmidhuber, 1997). One key

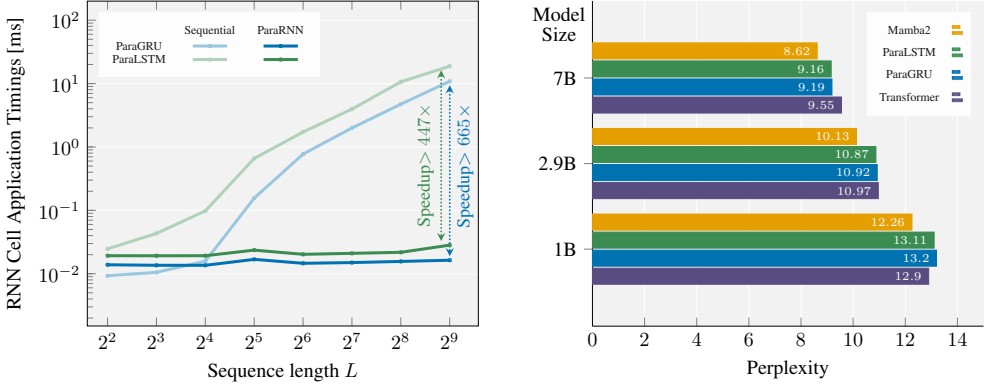

Figure 1: Our ParaRNN framework makes it possible to apply classical RNNs in parallel, dramatically speeding up their training, and allowing them to be used competitively for language modeling.

Codebase is available at `https://github.com/apple/ml-pararnn`.

reason behind the rapid adoption of Transformers lies in the efficiency of their application at training time: their core sequence mixer, the attention mechanism, can in fact be applied in parallel along the length of the input sequence. This effectively overcomes one main limitation of classical RNNs, whose application must be unrolled sequentially along the input sequence.

In more recent times, however, interest in RNNs has been rekindled, largely due to their reduced memory footprint and improved efficiency at inference time. In particular, recent advancements in *State Space Models* (SSMs) such as Mamba (Gu & Dao, 2023; Dao & Gu, 2024) have started gaining popularity and are imposing themselves as a potential alternative to Transformers, at least for small-to-mid-sized models (Zuo et al., 2024). To grant parallelization during training (and hence a comparable performance to attention), SSMs simplify the recurrence relationship at their core to one that is purely linear in the hidden state. This simplification enables leveraging associativity and parallel reduction operations to quickly compute the output of the application of an SSM to a whole input sequence in parallel along its length. Despite the success of modern SSMs, the linearity constraint remains a limitation which hinders their expressive power (Merrill et al., 2025; Cirone et al., 2025), and is dictated by necessity rather than choice.

With ParaRNN, we aim to overcome the constraint of linearity for SSMs, while unlocking parallelization also for nonlinear RNNs, thus enriching the space of viable options for sequence modeling.

The method acts by re-casting the sequential application of an RNN to an input sequence of length $L$ as the solution of a nonlinear system of $L$ equations. Said solution is recovered via Newton's method: the system is linearized, and its solution approximated iteratively. In light of the Markovian nature of classical RNNs, the resulting linearized system obeys a specific block bi-diagonal structure, which we leverage to recover the solution efficiently, using custom high-performance implementations of parallel reduction operations. We provide a detailed description of the method in Sec. 2.

For the sake of favoring the widespread adoption of this method among the research community, we make available the ParaRNN codebase. This takes care of automating the sequence-parallel application of an RNN cell, relying only on the definition of the RNN recurrence formula. The main structure of the framework is outlined in Sec. 4 and App. D.

Finally, we demonstrate the validity of ParaRNN in allowing the training of competitive RNNs for language modeling. To this end, in Sec. 3 we introduce two adaptations of the classical LSTM and GRU cells, tailored for efficient application within the ParaRNN framework. In Sec. 5, we compare their performance against the Transformer and Mamba architectures, proving their competitiveness both in terms of runtime and of effectiveness in language modeling tasks.

**Contributions**    To summarize, our main contributions can be listed as follows:

1. We adapt and develop efficient parallel reduction algorithms to enable training of nonlinear RNNs at unprecedented scales, expanding prior work on Newton-based RNN parallelization.

2. We demonstrate for the first time that classical nonlinear RNN models can be trained on language modeling tasks at scales of 7B parameters, and achieve competitive performance with Transformers, thus enriching the space of available architecture choices for LLMs.

3. We enable the exploration of new nonlinear RNN architectures for language modeling at scale, by introducing ParaRNN: a high-performance PyTorch+CUDA library that implements sequence-parallel training for any nonlinear cell from only the specification of its recurrence step, thereby automating the underlying parallelization complexity.

**Previous work**    There is a flourishing line of research around RNNs that can be applied in parallel (Bradbury et al., 2016; Martin & Cundy, 2018; Qin et al., 2023; Gu et al., 2020; 2022a;b; Orvieto et al., 2023), which culminates more recently in Mamba and Mamba2 (Gu & Dao, 2023; Dao & Gu, 2024), perhaps the most successful examples of modern SSMs. In a nutshell, SSMs revolve around a simplification of RNNs, that constrains the recurrence step to be purely *linear* in the state. The associativity of linear operations enables then the use of *parallel scan* (Hillis & Steele, 1986; Blelloch, 1990) to parallelize the RNN application. In principle, however, this approach to parallelization comes at the cost of expressivity: recent theoretical works warn in fact about the limitations of expressive power of linear SSMs (Merrill et al., 2025; Huang et al., 2025; Keiblinger, 2025). With our work, we effectively remove such linearity constraint.

Recent advantages in SSMs have also spawned a renewed interest in RNN adaptations. This is most noticeably represented by minGRU/minLSTM (Feng et al., 2024), and xLSTM (Beck et al., 2024). However, the RNNs proposed in Feng et al. (2024) still preserve a linear recurrence at their core, although wrapped in additional nonlinearities that however—we remark—do *not* interest its state. On the other hand, xLSTM proposes two alternative RNN cells: sLSTM and mLSTM. The first is purely linear, and parallelized similarly to Mamba; the second, being nonlinear, is still applied sequentially, without any attempt to parallelize its application sequence-wise: this comes with constraints on the size of the hidden states considered, otherwise its training-time application would become too costly. Another approach to scaling up nonlinear RNNs is outlined in Chaubard & Kochenderfer (2025), which mitigates the cost of backpropagation through time via gradient approximation, but still relies on efficiently computing sequential forward passes through the RNN.

The parallelization approach of ParaRNN takes inspiration from methods originally developed for time-parallelizing the solution of Ordinary Differential Equations (ODEs); see Gander (2015) for an overview. The specific idea of combining Newton's iterations and parallel reduction to solve for nonlinear sequential operations in parallel has been first introduced in Danieli et al. (2023), where it was applied to ResNets and Diffusion Models, but only hypothesized for language modeling and other sequence models. An application of this same approach has been investigated in Lim et al. (2024) and improved in Gonzalez et al. (2024), where a GRU cell is trained in parallel for time-series modeling. There, the authors consider the use of a *quasi*-Newton method, where a diagonal approximation of the full RNN Jacobian is employed for computational efficiency, but requires approximating the backward pass as well, which might affect training dynamics. In this work, we show empirically that directly adapting the GRU and LSTM cells definition allows us to achieve stable and scalable training, and successfully extend the methodology to more complex cells that were not previously tested.

Lastly, our work shares some looser connections with other approaches to speedup via parallelization of sequential operations. In Song et al. (2021); Santilli et al. (2023), Jacobi and Gauss-Seidel solvers are applied to accelerate ResNets applications, auto-regressive image generation, and Transformers inference; in Shih et al. (2023), Picard iterations are used to parallelize denoising steps in Diffusion.

## 2 PARALLEL APPLICATION OF NONLINEAR RNNs

In this work, we build upon the blueprint of Danieli et al. (2023); Lim et al. (2024), where the sequential application of an RNN is re-cast as a system of nonlinear equations which is then solved via Newton iterations, and adapt it to LLM-scale training. For completeness, in the following we briefly describe the main components of the method used.

**Newton's Method for Computing Nonlinear RNN Applications** Consider the application of an RNN to an input sequence $[\boldsymbol{x}_l]_{l=1}^L$ of length $L$, whose elements are themselves vectors $\boldsymbol{x}_l \in \mathbb{R}^{d_{in}}$. The RNN cell tracks the evolution of a hidden state $\boldsymbol{h}_l \in \mathbb{R}^{d_h}$ through its recurrent application:

$$\boldsymbol{h}_l = \boldsymbol{f}(\boldsymbol{h}_{l-1}, \boldsymbol{x}_l), \qquad \forall l = 1, \ldots, L, \tag{1}$$

starting from an all-zero initial state $\boldsymbol{h}_0 = \boldsymbol{0}$, with $\boldsymbol{f}$ determining the cell action, or recurrence step. By collating the relationships in (1) we can define a single system of $L$ equations in the unknowns $[\boldsymbol{h}_l]_{l=1}^L$. This has the form

$$\begin{cases} \boldsymbol{h}_1 - \boldsymbol{f}(\boldsymbol{0}, \boldsymbol{x}_1) = \boldsymbol{0} \\ \boldsymbol{h}_2 - \boldsymbol{f}(\boldsymbol{h}_1, \boldsymbol{x}_2) = \boldsymbol{0} \\ \vdots \\ \boldsymbol{h}_L - \boldsymbol{f}(\boldsymbol{h}_{L-1}, \boldsymbol{x}_L) = \boldsymbol{0}. \end{cases} \tag{2}$$

To solve this system, we rely on Newton's method (Ortega & Rheinboldt, 2000). The method recovers an approximate solution of (2) by iteratively solving its linearization, until convergence. More in detail, given the approximate solution at the $k$-th Newton iteration $[\boldsymbol{h}_l^k]_{l=1}^L$, this is updated

solving

$$
\begin{bmatrix}
I & & & \\
-J_{\boldsymbol{f}}|_{\boldsymbol{h}_1^k} & I & & \\
& \ddots & \ddots & \\
& & -J_{\boldsymbol{f}}|_{\boldsymbol{h}_{L-1}^k} & I
\end{bmatrix}
\begin{bmatrix}
\delta\boldsymbol{h}_1^k \\
\delta\boldsymbol{h}_2^k \\
\vdots \\
\delta\boldsymbol{h}_L^k
\end{bmatrix}
=
\begin{bmatrix}
\boldsymbol{f}(\boldsymbol{0},\boldsymbol{x}_1) - \boldsymbol{h}_1^k \\
\boldsymbol{f}(\boldsymbol{h}_1^k,\boldsymbol{x}_2) - \boldsymbol{h}_2^k \\
\vdots \\
\boldsymbol{f}(\boldsymbol{h}_{L-1}^k,\boldsymbol{x}_L) - \boldsymbol{h}_L^k
\end{bmatrix},
\tag{3}
$$

so that $[\boldsymbol{h}_l^{k+1}]_{l=1}^L = [\boldsymbol{h}_l^k + \delta\boldsymbol{h}_l^k]_{l=1}^L$. In (3), $J_{\boldsymbol{f}}|_{\boldsymbol{h}_l^k}$ denotes the Jacobian of the recurrence step (1) with respect to the hidden state $\boldsymbol{h}$, evaluated at $\boldsymbol{h}_l^k$. Pseudo-code for the method is given in Alg. 1a.

The second ingredient for effectively tackling the solution of (2) consists of an efficient solver for the highly-structured, block bi-diagonal linear system in (3). This is described in the following section.

**Parallel Reduction for All-at-once Solution of Block Bi-diagonal Linear Systems**    As shown in the previous section, thanks to Newton's method, we can re-cast the application of a nonlinear RNN[1] into the iterative solution of a linear system in the form (3). In principle, its solution $[\delta\boldsymbol{h}_l^k]_{l=1}^L$ could be explicitly recovered by forward substitution, sequentially unrolling the formula

$$
\delta\boldsymbol{h}_l^k = J_{\boldsymbol{f}}|_{\boldsymbol{h}_l^k} \delta\boldsymbol{h}_{l-1}^k + (\boldsymbol{f}(\boldsymbol{h}_{l-1}^k,\boldsymbol{x}_l) - \boldsymbol{h}_l^k), \qquad \forall l = 1,\ldots,L,
\tag{4}
$$

but this would defeat the purpose of parallelization. Instead, notice that (4) itself corresponds to the application of a linear RNN, where the Jacobians $J_{\boldsymbol{f}}|_{\boldsymbol{h}_l^k}$ cover the role of the state matrix, while the residuals $\boldsymbol{r}_l = \boldsymbol{f}(\boldsymbol{h}_{l-1}^k,\boldsymbol{x}_l) - \boldsymbol{h}_l^k$ cover that of the input (or forcing term). As such, the solution can be recovered explicitly by leveraging associativity of matrix multiplication when unrolling the recursion (4). We have in fact

$$
\delta\boldsymbol{h}_l^k = J_{\boldsymbol{f}}|_{\boldsymbol{h}_l^k} \delta\boldsymbol{h}_{l-1}^k + \boldsymbol{r}_l = J_{\boldsymbol{f}}|_{\boldsymbol{h}_l^k} \left( J_{\boldsymbol{f}}|_{\boldsymbol{h}_{l-1}^k} \delta\boldsymbol{h}_{l-2}^k + \boldsymbol{r}_{l-1} \right) + \boldsymbol{r}_l = \ldots = \sum_{s=1}^{l} \prod_{r=0}^{l-s-1} J_{\boldsymbol{f}}|_{\boldsymbol{h}_{l-r}^k} \boldsymbol{r}_s.
\tag{5}
$$

There is some redundancy in the terms appearing in the sum in (5) for different $l$'s: this is what allows specialized algorithms such as *prefix sum* (also known as parallel reduction, or parallel scan

---

| **Algorithm 1a:** Newton's method | **Algorithm 1b:** Parallel reduction |
|---|---|
| *// Initialization:* | **Input:** Jacobians $J_{\boldsymbol{f}}\|_{\boldsymbol{h}_l}$, residuals $\boldsymbol{r}_l$ |
| 1 Assemble initial guess $\boldsymbol{h}_l^0$; | *// Initialization:* |
| *// Newton iterations* | 1 $A_l = J_{\boldsymbol{f}}\|_{\boldsymbol{h}_l}$;    $\delta\boldsymbol{h}_l = \boldsymbol{r}_l$; |
| 2 **for** $k = 1$ **to** $N_{its}$ **do** | *// Reduction steps:* |
| *// Assemble system (3):* | 2 **for** $i = 0$ **to** $\log_2 L - 1$ **do** |
| 3    Compute Jacobians $J_{\boldsymbol{f}}\|_{\boldsymbol{h}_l^k}$; | 3    **parfor** $l = 2^i$ *to* $L$ **do** |
| 4    Compute residuals $\boldsymbol{r}_l^k$; | 4       $\delta\boldsymbol{h}_l \leftarrow \delta\boldsymbol{h}_l - A_l\delta\boldsymbol{h}_{l-2^i+1}$; |
| *// Solve in parallel* | 5       $A_l \leftarrow -A_l A_{l-2^i+1}$; |
| 5    $\boldsymbol{h}_l^k \leftarrow \boldsymbol{h}_l^{k-1} + \texttt{ParallelReduce}(J_{\boldsymbol{f}}\|_{\boldsymbol{h}_l^k}, \boldsymbol{r}_l^k)$; | 6 **return** $\delta\boldsymbol{h}_l$; |
| 6 **return** $\boldsymbol{h}_l^{N_{its}}$; | |

**Algorithm 1**: Pseudo-code for the parallel application of a nonlinear RNN cell: we use Newton (left) as outer solver for the corresponding nonlinear system (2), while parallel reduction (right) is used as inner solver for the linearized system (3).

---

[1]While the method presented here considers the general case where the cell action (1) is *nonlinear* in the state $\boldsymbol{h}_l$, it also naturally encompasses *linear* RNNs, such as Mamba and other SSMs. In these cases, we have

$$
\boldsymbol{f}_{\text{SSM}}(\boldsymbol{h}_{l-1},\boldsymbol{x}_l) = \boldsymbol{A}_l\boldsymbol{h}_{l-1} + \boldsymbol{B}_l\boldsymbol{x}_l,
$$

and the Jacobians in (3) reduce to the state transition matrices themselves, $J_{\boldsymbol{f}_{\text{SSM}}}\big|_{\boldsymbol{h}_{l-1}} \equiv \boldsymbol{A}_l$. Consequently, Newton's update reduces to the vanilla SSM application, and the target output is recovered in a single iteration.

(Hillis & Steele, 1986; Blelloch, 1990)) to compute the whole solution $[\delta \boldsymbol{h}_l^k]_{l=1}^L$ at once, in parallel, in $\mathcal{O}(\log_2 L)$ steps, rather than sequentially in $\mathcal{O}(L)$ steps. Pseudo-code for a naïve implementation of the algorithm is provided in Alg. 1b, but we refer to App. D.2 for the complete description of the more efficient CUDA implementation developed in this work.

**Parallel Backward Pass Through RNN Applications**  Unlike the forward pass, which for general RNNs requires the solution of a nonlinear system, the backward pass through an RNN cell is an inherently linear operation. As such, we do not need to rely on Newton iterations, and the necessary gradients can be recovered directly in a single parallel reduction step. More in detail, given the partial derivatives of the loss function $\mathcal{L}$ with respect to the cell hidden states $[\partial_{\boldsymbol{h}_l} \mathcal{L}]_{l=1}^L$, computing the full gradients amounts to unrolling the backward recurrence (starting from $\nabla_{\boldsymbol{h}_L} \mathcal{L} = \partial_{\boldsymbol{h}_L} \mathcal{L}$)

$$\nabla_{\boldsymbol{h}_{l-1}} \mathcal{L} = J_{\boldsymbol{f}}\big|_{\boldsymbol{h}_{l-1}}^\top \nabla_{\boldsymbol{h}_l} \mathcal{L} + \partial_{\boldsymbol{h}_{l-1}} \mathcal{L}, \qquad \forall l = L, \dots, 1. \tag{6}$$

Notice this shares the same structure of (5), except it involves transposes of Jacobians, and it unrolls backwards. This notwithstanding, (6) can be solved using the same parallel reduction algorithm Alg. 1b, with minimal modifications.

## 2.1 LIMITATIONS

While the ParaRNN framework described in this section is general enough to be applied to virtually any Markovian RNN, it presents two major limitations that must be kept into account when deploying it in practical applications.

**Newton's Convergence**  Our method relies on the convergence of Newton's iterations to recover the result from the RNN cell application. Gonzalez et al. (2024) showed that, for systems like (2), convergence in $L$ Newton steps is guaranteed. However, such a slow convergence is not practical: it would break any advantage from parallelization, making a vanilla sequential application of the RNN cell more efficient. In practice, we need convergence in a small $\mathcal{O}(1)$ number of iterations. Fortunately, for the RNN considered in this work, *three* iterations proved to be sufficient in all cases, but this must be verified for newly defined RNN wanting to leverage ParaRNN. For additional theoretical guarantees in this sense, we refer to the recent work in Gonzalez et al. (2025), while in App. A we report more information regarding the Newton solver setup and convergence for our case.

**Computational Efficiency**  While ParaRNN allows to attain sequence-parallelizability, it does so at the cost of additional overhead computations, which would not be necessary in the sequential case. This is a common trade-off in the design of parallel algorithms. In our case, the brunt of this overhead is associated with the assembly and manipulation of $J_{\boldsymbol{f}}$ terms in (5). In fact, if we were to naïvely consider RNNs with dense Jacobians, their memory requirement would scale as $\mathcal{O}(L d_h^2)$, and each pair-wise multiplication (line 5 of Alg. 1b) would incur a computational cost of $\mathcal{O}(d_h^3)$, making it *de-facto* unfeasible. In practice, then, one should inject some additional structure in the Jacobians definition, so to make these operations more tractable. Notice that a similar problem arises when dealing with SSMs: to overcome this, e.g. in S4D (Gu et al., 2022b) the authors impose a diagonal structure on the SSM state matrix, to render the application of the prefix sum algorithm computationally feasible. This assumption is inherited by the Mamba model (Gu & Dao, 2023), for analogous considerations. Similarly, Gonzalez et al. (2024) opt to use a *quasi*-Newton method, where the RNN Jacobian is approximated with its diagonal, again rendering the resulting matrix multiplications leaner. In our case, we follow an approach similar to Mamba, as outlined next.

## 3 ADAPTING GRU AND LSTM FOR PARARNN

As mentioned in Sec. 2.1, the main obstacle to the efficient application of the prefix scan algorithm to (5) lies in the computation of its product terms $\prod_{r=0}^{l-s-1} J_{\boldsymbol{f}}\big|_{\boldsymbol{h}_{l-r}^k}$. Rather than considering Jacobians approximations as in Gonzalez et al. (2024), in our approach we choose instead to directly tune the RNN definition to yield Jacobians with a simplified structure. The adapted GRU and LSTM cells are described next.

**RNN Cell Definitions**  In defining the main RNNs used in our work, we start from two of the most broadly established architectures available in the literature: the *Gated Recurrent Unit* (GRU) (Cho et al., 2014) and the *Long-Short Term Memory* (LSTM) (Hochreiter & Schmidhuber, 1997) cells. For the latter, we consider its variant equipped with peephole connections and combining input and forget gates (Greff et al., 2017). Their actions are defined as, respectively:

$$
\text{GRU} \begin{cases} \boldsymbol{z}_l = \sigma_g \left( A_z \boldsymbol{h}_{l-1} + B_z \boldsymbol{x}_l + \boldsymbol{b}_z \right) \\ \boldsymbol{r}_l = \sigma_g \left( A_r \boldsymbol{h}_{l-1} + B_r \boldsymbol{x}_l + \boldsymbol{b}_r \right) \\ \boldsymbol{c}_l = \sigma_h \left( A_c (\boldsymbol{h}_{l-1} \odot \boldsymbol{r}_l) + B_c \boldsymbol{x}_l + \boldsymbol{b}_c \right) \\ \boxed{\boldsymbol{h}_l = (\boldsymbol{1} - \boldsymbol{z}_l) \odot \boldsymbol{h}_{l-1} + \boldsymbol{z}_l \odot \boldsymbol{c}_l} \end{cases} , \qquad \text{LSTM} \begin{cases} \boldsymbol{f}_l = \sigma_g \left( A_f \boldsymbol{h}_{l-1} + B_f \boldsymbol{x}_l + C_f \boldsymbol{c}_{l-1} + \boldsymbol{b}_f \right) \\ \boldsymbol{z}_l = \sigma_z \left( A_z \boldsymbol{h}_{l-1} + B_z \boldsymbol{x}_l + \boldsymbol{b}_z \right) \\ \boxed{\boldsymbol{c}_l = \boldsymbol{f}_l \odot \boldsymbol{c}_{l-1} + (\boldsymbol{1} - \boldsymbol{f}_l) \odot \boldsymbol{z}_l} \\ \boldsymbol{o}_l = \sigma_g \left( A_o \boldsymbol{h}_{l-1} + B_o \boldsymbol{x}_l + C_o \boldsymbol{c}_l + \boldsymbol{b}_o \right) \\ \boxed{\boldsymbol{h}_l = \boldsymbol{o}_l \odot \sigma_h(\boldsymbol{c}_l)} \end{cases}
$$

$$(7a) \qquad\qquad (7b)$$

for $l = 1 \dots L$. The highlighted equations represent the actual hidden state of the two cells (that is, the variables involved in the recurrence step), while the other ones are auxiliary temporary variables needed to perform the update itself. Notice in particular that for LSTM the hidden state is given by the collation of two variables: $\hat{\boldsymbol{h}}_l = [\boldsymbol{c}_l^\top, \boldsymbol{h}_l^\top]^\top \in \mathbb{R}^{2d_h}$.

**Cell Jacobians**  To apply the ParaRNN machinery described in Sec. 2 to the cells in (7), we need to compute the Jacobians of their recurrence step with respect to the hidden state. These can be recovered explicitly with some algebraic operations, and come in the form, respectively:

$$
J_{\text{GRU}} = \text{diag}(\boldsymbol{1} - \boldsymbol{z}_l) + \text{diag}((\boldsymbol{c}_l - \boldsymbol{h}_{l-1}) \odot \sigma_g'(\hat{\boldsymbol{z}}_l)) A_z
$$
$$
+ \text{diag}(\boldsymbol{z}_l \odot \sigma_h'(\hat{\boldsymbol{c}}_l)) A_c (\text{diag}(\boldsymbol{r}_l) + \text{diag}(\boldsymbol{h}_{l-1} \odot \sigma_g'(\hat{\boldsymbol{r}}_l)) A_r), \quad \text{and} \qquad (8a)
$$

$$
J_{\text{LSTM}} = \left[ \begin{array}{c|c} J_{\boldsymbol{cc}} & J_{\boldsymbol{ch}} \\ \hline J_{\boldsymbol{hc}} & J_{\boldsymbol{hh}} \end{array} \right], \quad \text{with} \quad \begin{cases} J_{\boldsymbol{cc}} = \text{diag}(\boldsymbol{f}_l) + \text{diag}((\boldsymbol{c}_{l-1} - \boldsymbol{z}_l) \odot \sigma_g'(\hat{\boldsymbol{f}}_l)) C_f \\ J_{\boldsymbol{ch}} = \text{diag}((\boldsymbol{c}_{l-1} - \boldsymbol{z}_l) \odot \sigma_g'((\hat{\boldsymbol{f}}_l))) A_f \\ \qquad + \text{diag}((\boldsymbol{1} - \boldsymbol{f}_l) \odot \sigma_z'(\hat{\boldsymbol{z}}_l)) A_z \\ J_{\boldsymbol{hc}} = (\text{diag}(\sigma_h(\boldsymbol{c}_l) \odot \sigma_g'(\hat{\boldsymbol{o}}_l)) C_o \\ \qquad + \text{diag}(\boldsymbol{o}_l \odot \sigma_h'(\boldsymbol{c}_l))) J_{\boldsymbol{cc}} \\ J_{\boldsymbol{hh}} = \text{diag}(\sigma_h(\boldsymbol{c}_l) \odot \sigma_g'(\hat{\boldsymbol{o}}_l))(A_o + C_o J_{\boldsymbol{ch}}) \\ \qquad + \text{diag}(\boldsymbol{o}_l \odot \sigma_h'(\boldsymbol{c}_l)) J_{\boldsymbol{ch}}, \end{cases} \qquad (8b)
$$

where $\text{diag}(\boldsymbol{v})$ denotes the diagonal matrix with the elements from $\boldsymbol{v}$ as its diagonal, $\sigma_*'$ is the derivative of $\sigma_*$, and for a given temporary variable $\boldsymbol{v}$, $\hat{\boldsymbol{v}}$ denotes its value before the nonlinearity application. Notice that the LSTM Jacobian features a $2 \times 2$-block structure, directly stemming from the fact that the hidden state is split into two variables—so that each sub-Jacobian $J_{**}$ indicates the partial derivative $\partial(*)_l / \partial(\star)_{l-1}$.

**Jacobians Structure Simplification**  Our goal is to make Jacobians in (8) leaner to store and to multiply together. Notice that the overall structure of these Jacobians is ultimately linked to that of the various state and peephole-connection matrices $A_*, C_\star$, since all other matrices involved are purely diagonal. An apt simplification consists then in picking these matrices, too, to be diagonal; in other words, we substitute in (7):

$$
A_* = \text{diag}(\boldsymbol{a}_*), \quad C_\star = \text{diag}(\boldsymbol{c}_\star), \qquad \text{for some learnable parameters} \qquad \boldsymbol{a}_*, \boldsymbol{c}_\star \in \mathbb{R}^{d_h}. \quad (9)
$$

This reduces $J_{\text{GRU}}$ in (8) to a diagonal matrix, and further simplifies the $2 \times 2$-block structure of $J_{\text{LSTM}}$ to one with diagonal blocks. In both cases, the Jacobians thus simplified occupy only $\mathcal{O}(Ld_h)$ memory overall (we verify this also empirically in App. B.3), and each pairwise multiplication can be performed with $\mathcal{O}(d_h)$ complexity, in a perfectly parallelizable fashion over the hidden state components.

While this simplification allows us to satisfy the requirements (outlined in Sec. 2.1) to apply parallel scan efficiently, we point out that it has the effect of inhibiting any mixing of the hidden state components *within* the RNN cell: indeed, with (9) we are effectively reducing a single $d_h$-dimensional RNN cell to $d_h$ independent 1-dimensional ones. This has an impact on expressivity, as we explore in Sec. 5.2. We stress however that this choice of structure is not an inherent requirement for

ParaRNN. Alternative Jacobian shapes can be accommodated by the framework, so long as they allow for efficient parallel reduction (see Walker et al. (2025) for an excellent overview of feasible structure choices and their implication on expressivity, or Yang et al. (2025); Siems et al. (2025) for applications with Householder matrices). We adopt the diagonal constraint here merely to keep our work contained, and to align with standard practices in modern SSMs: Mamba employs a diagonal state matrix (further simplified to a single scalar parameter in Mamba2), and diagonalized RNN cells appear in works ranging from Lei et al. (2018) (to boost efficiency in the *sequential* cell application) to more recent efforts like Farsang et al. (2025) (targeting parallelizability, as we do). This notwithstanding, exploring alternative structures within ParaRNN remains an exciting avenue for future work.

## 4 Efficient implementation of parallel reduction

As part of the main contributions of this work, we make available ParaRNN, a fully modular, PyTorch+CUDA-based software framework allowing to readily parallelize the application of virtually *any* user-defined RNN cell. The user needs only to prescribe the recurrence step defining the cell action (including any parameters necessary to its definition), and the ParaRNN framework takes care of automatically assembling the resulting all-at-once system. Particularly, it leverages `autograd` to assemble the Jacobians required in (3), and then proceeds to efficiently solve the system via Newton's method and parallel reduction (Alg. 1), parallelizing the RNN cell application.

In ParaRNN we provide three different implementations of the parallel solver, allowing users to trade off ease-of-use against performance:

1. **Pure PyTorch** A reference implementation of parallel reduction which only relies on native PyTorch operations, and is mainly thought for prototyping, debugging, and exploring new RNN cell definitions. While not directly optimized for performance, this version provides full automatic differentiation support and seamless integration with existing PyTorch models. By manually providing the explicit formula for the Jacobians assembly, the user can still choose to boost performance by overriding the default `autograd`-based ones.

2. **CUDA-Accelerated Parallel Reduction** A performant implementation featuring a custom CUDA kernel for the parallel reduction solver, specialized for Jacobians with diagonal or $N \times N$-block-diagonal structure as in (8). Jacobian assembly remains in PyTorch, preserving automatic differentiation capabilities while accelerating the computational bottleneck. The CUDA solver employs a hardware-aware, hybrid algorithm combining forward substitution and parallel reduction at distinct levels in the GPU hierarchy (*thread*, *warp*, *block* and *device*, see NVIDIA (2025)). The goal is to maximize register utilization and take full advantage of efficient warp-level directories, while minimizing shared and global memory traffic: we refer to App. D.2 for full implementation details.

3. **Fully-fused** Our most performant implementation where the whole Newton routine, including iterative Jacobian assembly and parallel solution, is fused within a single CUDA kernel. This eliminates intermediate memory traffic and overhead from kernel function calls, although it requires users to provide a CUDA implementation of the RNN cell. Nonetheless, we still preserve a modular approach, so that the user only needs to define those operations which directly pertain to the cell itself (namely, recurrence step and Jacobian assembly: see App. D.3). Our code provides a reference implementation for the ParaGRU and ParaLSTM cells described in Sec. 3.

## 5 Results

The experiments reported in this section aim at evaluating the competitiveness of ParaRNN in Language Modeling tasks, particularly when applied to the ParaGRU and ParaLSTM cells. To this purpose, we focus on two main metrics. First, in Sec. 5.1, we investigate the computational cost (primarily in terms of wall-time) associated with utilizing ParaRNN: our goal is to demonstrate that, by unlocking sequence-parallelizability, ParaRNN allows to (i) train nonlinear-RNN-based models with a runtime comparable to that of Transformer and Mamba architectures; and (ii) achieve inference speed much faster than Transformer, and similar to Mamba. Secondly, in Sec. 5.2 we analyze the overall performance of ParaGRU and ParaLSTM as language models, and show that they can achieve competitive perplexity and downstream task performance.

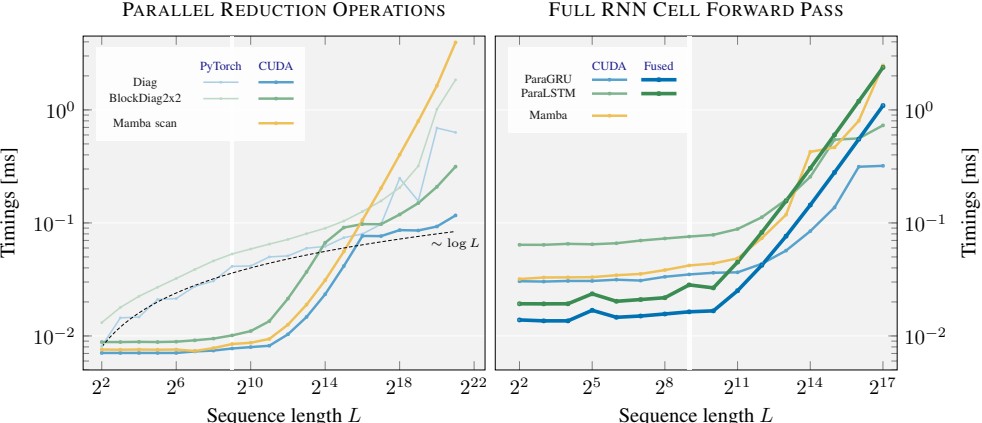

Figure 2: Timing results for the parallel reduction operations Alg. 1b (left), and the full parallel RNN application Alg. 1a (right), applied to sequences of varying length. Colors refer to different variants of the solvers: for diagonal Jacobians (blue, used by ParaGRU) and 2x2 block-diagonal ones (green, used by ParaLSTM). Heavier lines refer to progressively more efficient implementations: in PyTorch, in CUDA, and (for the RNN forward pass) with the fully-fused CUDA kernel. For reference, equivalent timing results for Mamba are also included (yellow). Actual speedup measurements reported in the main text refer to $L = 2^9$ (highlighted).

## 5.1 IMPLEMENTATION PERFORMANCE

**Competitive Runtime** Here we report profiling results for the main functions involved in the application of ParaRNN. First, in Fig. 2 (left) we time our different implementations of the parallel reduction operation Alg. 1b. Notably, our CUDA implementation for diagonal Jacobians slightly improves on the equivalent `parallel_scan` shipped with Mamba ($\sim 1.1\times$ speedup at $L = 2^9$); the one for block-diagonal Jacobians is instead slower ($\sim 0.84\times$ slowdown at $L = 2^9$)—as expected, being it more memory and computationally intensive. Either case, the brunt of the kernel cost is associated with the memory reading overhead, rather than the computation itself, as indicated by the lines being mostly flat until $L \sim 2^{11}$. The inherent complexity of the operation is instead more visible from the profiling of the PyTorch implementation: this closely follows a logarithmic growth, before collapsing to linear (from $L \gtrsim 2^{18}$), when the GPU is close to capacity and operations must be sequentialized. The profiling curves also mirror the three regimes of our parallel scan implementation described in App. D.2: (i) the sequence fits in a single GPU block and is analyzed in parallel within it ($L \lesssim 2^{10}$); (ii) the sequence is analyzed in parallel within each block, but sequentially over a few blocks ($2^{10} \lesssim L \lesssim 2^{16}$, where the curve rises to a linear regime); and (iii) the sequence is analyzed in parallel across blocks ($L \gtrsim 2^{16}$, where we observe the curve flattening again to a log). By contrast, Mamba's implementation collapses to sequential across blocks for $L \gtrsim 2^{10}$.

Similar observations on relative performance hold also when we consider the complete parallel application of the RNN cell forward pass—that is, the full Newton routine Alg. 1 instead of the sole parallel reduction operation Alg. 1b. Profiling results for this setting are reported in Fig. 2 (right). Already our implementation combining Jacobians assembly in PyTorch and parallel reduction in CUDA provides runtimes equivalent to the Mamba SSM application: this is basically comparable to applying ParaGRU (we attain a $\sim 1.2\times$ speedup for $L = 2^9$), while ParaLSTM still lags behind (with a $\sim 0.5\times$ slowdown), again following expectations. However, it is by fully-fusing the whole Newton routine in CUDA that we achieve the largest speedups with respect to Mamba: of $\sim 2.6\times$ and $\sim 1.5\times$ for ParaGRU and ParaLSTM respectively, at $L = 2^9$.

**Fast Inference** For inference, we observe a runtime similar to that of Mamba, as shown in Fig. 3: at regime, we attain a throughput of $\sim 37 - 38$ tokens per second, versus the $\sim 28$ of Mamba. Notice moreover that, unlike for Transformers, for RNNs the time necessary to autoregressively produce the next token in the output *does not grow with sequence length*.

Overall, the measurements reported in this section support our claim that ParaRNN represents a viable option for accelerating the application of RNNs, with runtime competitive with (and often beating) Mamba. We refer to App. B for additional timing results and more details on the setup for the experiments considered here.

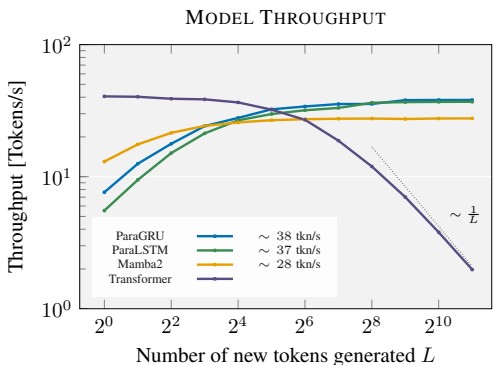

Figure 3: Tokens throughput at generation for ParaGRU (blue), ParaLSTM (green), Mamba (yellow), and Transformers (purple).

## 5.2 LANGUAGE MODELING PERFORMANCE

Having established runtime parity between Mamba and our RNN cells, we now analyze a different axis of ParaGRU and ParaLSTM effectiveness: their performance as actual language models. Our goal is to show that, with sequentiality constraints removed and training at scale enabled, even classical RNNs can achieve competitive performance in language modeling tasks.

**The Importance of Nonlinearities**    We begin by evaluating ParaGRU and ParaLSTM on synthetic tasks that isolate fundamental capabilities required for language modeling. We focus on retrieval and state-tracking tasks (Arora et al., 2023), where linear SSMs like Mamba are known to struggle (Merrill et al., 2025)—even though minor modifications to the eigenvalues distribution of their state matrix can help, at least for specific tasks (Grazzi et al., 2025). The goal of this section is to establish the critical motivation behind our work: *nonlinear* RNNs provide computational expressivity that *linear* RNNs fundamentally lack, and hence merit continued development.

To directly assess the expressivity of the recurrent cell in isolation, we use single-layer architectures consisting only of an embedding layer, normalization, the RNN cell considered (or Attention, for comparison), and a linear layer—see a detailed description in App. C.2. As shown in Tab. 1, ParaGRU and ParaLSTM can solve perfectly the majority of tasks considered, with near-perfect solutions recovered for Cycle Navigation and Modular Arithmetic tasks. However, both cells (as well as Mamba) struggle on the more complex $A_5$ task. This limitation is directly attributable to the diagonal structure adopted in our work (see also Merrill et al. (2025)), as evidenced by the perfect scores achieved by vanilla LSTM and GRU cells with full Jacobians—also reported in Tab. 1 for reference. Most importantly, though, the nonlinear cells consistently outperform the linear Mamba2 across all tasks considered, highlighting the critical role of nonlinearities in enhancing expressivity.

**Matching Transformers Without Attention**    Moving beyond synthetic tasks, we now evaluate our cells on language modeling to assess their effectiveness. To this end, we train four models: our ParaGRU and ParaLSTM presented in Sec. 3, plus a Transformer with the DCLM architecture (Li et al., 2024) and Mamba2 (Dao & Gu, 2024) as baselines. For each architecture, we consider four different model sizes (of 400M, 1B, 2.9B, and 7B parameters), and follow Chinchilla-optimal (Hoffmann et al., 2022) scaling for their training setup, using SlimPajama (Soboleva et al., 2023) without the Books3 split as training dataset: a complete description of the training procedure, including

Table 1: Single-layer accuracy on synthetic tasks. $()^{\dagger}$ denotes results computed training only the RNN cell (see App. C.2 for details).

| Model | Keep-n$^{\text{th}\dagger}$ $n = 5$ $\lvert V \rvert = 128$ | Parity$^{\dagger}$ $\lvert V \rvert = 2$ | MQAR $\kappa = 2$ $\lvert V \rvert = 128$ | $k$-hop $k = 1$ $\lvert V \rvert = 10$ | $k$-hop $k = 2$ $\lvert V \rvert = 5$ | Cycle Nav $k = 1$ $C = 6$ | Mod Arith (w/o brackets) $\lvert V \rvert = 10$ | Copy Mem $M = 20$ $\lvert V \rvert = 10$ | $A_5$ |
|---|---|---|---|---|---|---|---|---|---|
| LSTM | 100% | 100% | 100% | 100% | 100% | 100% | 100% | 100% | 100% |
| GRU | 100% | 100% | 100% | 100% | 100% | 100% | 100% | 100% | 100% |
| ParaLSTM | 100% | 100% | 100% | 100% | 100% | 95% | 94% | 67% | 38% |
| ParaGRU | 100% | 100% | 100% | 100% | 100% | 97% | 90% | 63% | 40% |
| Mamba2 | 1% | 52% | 100% | 100% | 98% | 57% | 44% | 55% | 36% |
| Transformer | 100% | 53% | 100% | 82% | 78% | 100% | 57% | 100% | 28% |

Table 2: Perplexity, parameters count, and evaluation scores on `lm-eval-harness` (Gao et al., 2021) tasks, for 7B models. Accuracies in percentages. Shot counts are reported in brackets.

| Model | #params | ↓ PPL | ↑ Arc-C (25) (3) | ↑ HSwag (10) (0) | ↑ OBQA (10) (0) | ↑ WinoG (5) (0) | ↑ PiQA (0) | ↑ MMLU (0) |
|---|---|---|---|---|---|---|---|---|
| Mamba2 | 6.96B | 8.62 | 40.02 39.59 | 69.78 69.68 | 42.20 42.20 | 65.19 63.77 | 76.66 | 26.61 |
| ParaLSTM | 6.76B | 9.16 | 37.46 36.52 | 62.47 62.85 | 42.20 38.80 | 57.70 59.12 | 75.19 | 25.31 |
| ParaGRU | 6.76B | 9.19 | 39.68 36.77 | 65.85 65.75 | 42.20 40.40 | 61.40 59.83 | 76.66 | 25.29 |
| Transformer | 6.89B | 9.55 | 34.30 33.36 | 62.98 62.20 | 40.00 37.20 | 61.48 60.85 | 74.97 | 23.12 |

hyperparameters configuration and training times is reported in App. C.1. In designing our RNN models, we build upon the DCLM Transformer backbone, replacing attention with our RNN cells, but including also the causal convolution and gated residual layers from Mamba: detailed architecture specifications are listed in App. C.1, and a schematic of the RNN block is illustrated in Fig. 10.

Table 2 presents the downstream task performance of our 7B-parameters trained models on reference tasks from the `lm-eval-harness` suite (Gao et al., 2021) (see Tab. 6 for complete results across scales). Results reveal a consistent ordering, in line with the perplexities reported in Fig. 1: Mamba2 achieves the strongest performance, followed by our ParaGRU and ParaLSTM models, with the DCLM Transformer trailing behind. To properly contextualize these results, we remark that our goal is not to propose a novel, more performant RNN, but to prove that also classical nonlinear RNNs can be trained efficiently at scale through parallelization, achieving performance competitive with modern language models. Ultimately, we are showing that the training inefficiencies that historically sidelined classic RNNs can be overcome, reviving them as viable options for language modeling.

## 6 CONCLUSION AND FUTURE WORK

In this work, we demonstrate that the training-parallelization barrier for nonlinear RNNs can be effectively overcome even at scale, the linearity constraint no longer precluding additional explorations in efficient sequence modeling. Our empirical results show that even classical RNNs, when trained with ParaRNN, can achieve performance competitive with modern architectures—crucially, without sacrificing training efficiency. This parity represents an important first step: it establishes that nonlinear RNNs are again a viable alternative in the modern landscape of sequence modeling.

In addition, the ParaRNN codebase makes this alternative readily available to practitioners, while delivering performance that matches or exceeds existing parallel implementations. By proving that classical RNNs can compete on equal footing with current architectures once computational barriers are removed, we aim for this work to further reignite exploration into nonlinear recurrent models.

## ACKNOWLEDGMENTS

We would like to extend our thanks to: Hadi Pour Ansari, for all the help during the reviewing process of this manuscript; Eeshan Gunesh Dhekane and Jagrit Digani, for the extremely valuable feedback, particularly regarding the CUDA kernels implementation; and Nick Apostoloff and Jerremy Holland, for their support throughout this project. Finally, we are grateful to the ICLR Reviewers and Area Chair for their recommendations and contribution in increasing the quality of this work.

## REPRODUCIBILITY STATEMENT

To ensure reproducibility, all our models were trained using public datasets and open source libraries. The code to reproduce the algorithms described in this work is available on GitHub at `https://github.com/apple/ml-pararnn/`. In addition, this document and its appendices contains all the necessary details to reproduce the experiments. We also include pseudo-code for the Newton and Parallel reduction algorithms in Alg. 1, as well as details and samples of the software package implementation in App. D, and hyperparameter details in App. B and C.1.

## ETHICS STATEMENT

Our work primarily focuses on advancing the computational efficiency and scalability of recurrent neural networks through novel algorithmic and architectural innovations. As such, it is a foundational technical contribution that does not directly involve the collection, processing, or deployment of sensitive user data, nor does it present immediate ethical concerns related to bias, privacy, or misuse in its core methodology. Furthermore, in our efforts to promote responsible AI development, we have taken proactive steps regarding data integrity and copyright, such as the removal of copyrighted material like Books3 from SlimPajama. We are committed to responsible research and development, and by open-sourcing the ParaRNN codebase, we aim to encourage transparent and collaborative progress in the field of generative modeling.

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

## A  NEWTON CONVERGENCE FOR PARAGRU AND PARALSTM

**Summary**

In this section, we empirically verify that ParaRNN's Newton algorithm is stable and efficient when applied to ParaGRU and ParaLSTM, consistently converging in 3 iterations, regardless of RNN cell used and stage of training considered.

In Sec. 2.1 we hinted at the necessity of the Newton's method Alg. 1a to converge quickly for ParaRNN to be appealing as a parallelization method for RNN cells applications. In practice, solving (2) in parallel comes at a computational complexity of $\mathcal{O}(N_{\text{its}} \log_2 L)$, if $N_{\text{its}}$ is the number of Newton iterations required for convergence: keeping $N_{\text{its}} \sim \mathcal{O}(1)$ is then paramount. Ultimately, the convergence behavior of Newton's method will depend on the actual definition of the cell action $\boldsymbol{f}$ in (1), and providing a comprehensive theoretical study of the properties of $\boldsymbol{f}$ necessary for fast convergence is beyond the scope of this project and a topic for future research–although we refer to Gonzalez et al. (2025) for the relevance of Lyapunov stability in this sense. Here, we limit ourselves to demonstrating empirically that, when applied to the ParaGRU and ParaLSTM models considered in this work, Newton's method consistently converges around the third iteration.

To this end, in Fig. 4 we examine the convergence behavior of Newton's method across different stages of model training and for varying sequence lengths. By default, in all forward applications of any RNN cell (including during the training of the models described in Sec. 5.2), we consider as an initial guess $\boldsymbol{h}^0$ for the Newton iterations the quantity

$$\boldsymbol{h}_l^0 = \boldsymbol{f}(\boldsymbol{0}, \boldsymbol{x}_l), \qquad \forall l = 1, \dots, L. \tag{10}$$

That is, for each input $\boldsymbol{x}_l$, we apply the cell action considering an all-zero previous hidden state $\boldsymbol{0}$. Notice this is computed in a perfectly parallel fashion along the sequence length $L$.

Figure 4 considers two complementary regimes, to show that Newton iterations are stable throughout the training procedure. On the left we show residual norm convergence for freshly-initialized RNN cells (see App. C.1 for the initialization scheme considered), while on the right we consider RNN cells extracted from the last layer of the fully-trained 400M-parameter models described in App. C.1. As evident from the graphs, the method exhibits rapid convergence: the residual drops to machine precision within 3-4 iterations, regardless of RNN cell considered or training stage, and convergence rate is only very marginally affected by an increase in the sequence length. For freshly initialized ParaLSTM, the residual at the very first iteration increases, indicating potential marginal

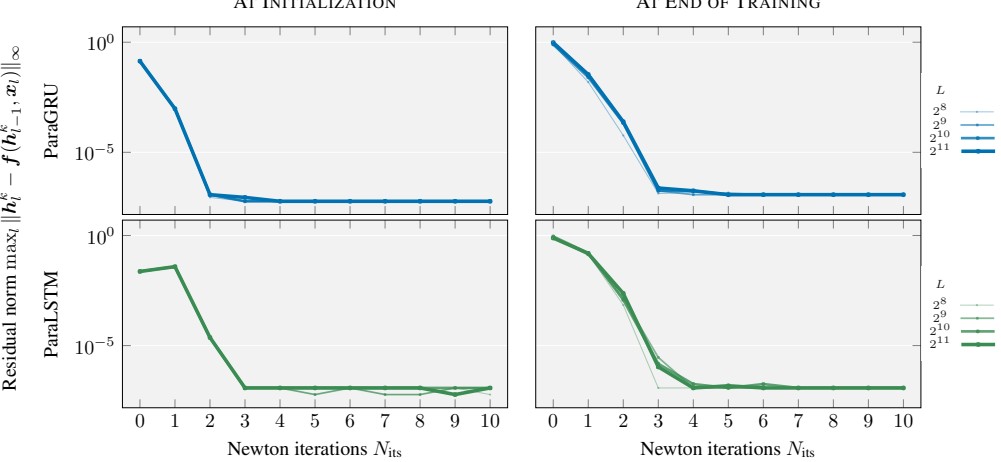

Figure 4: Newton's method convergence behavior for ParaGRU (top) and ParaLSTM (bottom) cells with different input sequence lengths $L$, for freshly initialized RNN cells (left), and for the RNN cells in the last layer of our trained 400M models (right). Inputs are randomly selected from the evaluation datasets, and the residuals reported are the max on batches of size 8.

for improvement in defining an initial guess for Newton's method. For ParaGRU, just 2 iterations suffice at initialization, growing to 3 at the end of training. These results guided our choice of $N_{\text{its}} = 3$ for the experiments in Sec. 5.2.

This consistent behavior across both initialization and trained states suggests that the nonlinearity structure of the ParaGRU and ParaLSTM cells lends itself well to a solution via Newton's method, but—we remark—this is not in general guaranteed for every choice of RNN cell.

## B  DETAILS ON PROFILING RESULTS

**Summary**

In this section we expand the results in Sec. 5.1 reporting timings for all application modalities of the RNN cells, including sequential and pure PyTorch implementations. The fully-fused forward pass applications of ParaRNN are always faster or comparable to Mamba for all the sequence lengths tested, and beat Mamba consistently when the backward pass is also taken into account. We also present an ablation over the `chunk_size` hyperparameter, controlling thread-level workload in our CUDA implementations. The resulting optimal values are model- and application mode-specific, with values $1 \sim 4$ generally yielding best runtimes.

We additionally confirm the throughput results in Fig. 3 hold across all scales considered, and that the parallel application of the RNN cells requires $\mathcal{O}(L\, d_h)$ memory.

**Experiments Setup**  All timings in Sec. 5.1 are collected on an NVIDIA A100 GPU. For Fig. 2, profiling is performed by tracking CUDA `Events` (NVIDIA, 2025, Sec. 6.2.8.8.2.). The profiler first warms-up the GPU with 20 dry runs of the target kernel to measure, before proceeding to collecting measurements from 100 runs processing random input sequences of a given length $L$. In the figures we report the minimum[2] timings for each sequence length considered. To provide a fairer comparison against production-ready code, all PyTorch baseline measurements (including Mamba) are performed using `torch.compile()`-optimized code, rather than eager execution mode, as this represents the standard deployment configuration for real-world applications. Also for the sake of fairness, for the Mamba baselines in Fig. 2 we consider: on the left, the application of the sole `parallel_scan` routine at the core of Mamba's SSM application; on the right, the application of a *simplified* Mamba module, stripped of its gate and convolution components (Gu & Dao, 2023)—in other words, we track only those operations necessary for the assembly of the SSM parameters and the application of the SSM cell. Finally, full RNN cell application timings consider the default setup with $N_{\text{its}} = 3$ Newton iterations.

For the throughput results in Fig. 3, we consider the 1B models discussed in Sec. 5.2. To profile, we simply use Python's `timeit` module, and report the minimum timings over 5 repetitions on an NVIDIA A100 GPU. The reduced number of repetitions in this case is mainly due to the lengthy generation times required by the Transformer; nonetheless, the reported curves appear smooth, so we consider them sufficient to provide an accurate measure.

### B.1  ADDITIONAL TIMING RESULTS

**Forward / Backward Pass Varying Application Modality**  In Fig. 5 we expand on the results reported in Fig. 2, including timings for the sequential application of the RNN cells, as well as from the pure PyTorch implementation. The former results were also used to compute the speedup measurements reported in Fig. 1. Overall, the curves behave as expected: applying the RNN cells sequentially becomes quickly computationally unfeasible due to its linear-growth complexity, and moving to progressively more efficient implementations (from sequential, to parallel in PyTorch, to computing the parallel reduction operation in CUDA, to a fully-fused CUDA kernel for the whole Newton routine) lowers the overall runtime for the RNN application. Perhaps surprisingly, the curve for the PyTorch implementation of ParaGRU in Fig. 5 (left) is ragged. We speculate that this is due to the effect of `torch.compile()`, which for this cell is able to unlock certain optimizations for specific sequence lengths but not for others. Notice also that the CUDA implementation of ParaGRU surpasses the fully-fused one for $L > 2^{12}$. To explain this unexpected result we need to refer to the specifics of the CUDA kernel implementation outlined in App. D.2. Fully-fused kernels exert a larger pressure on the GPU registers, in light of the larger number of computations they must

---

[2]We report the *minimum* following established benchmarking practices in high-performance computing: it represents a cleaner metric, and a more faithful indication of the actual performance of a kernel operating under ideal conditions. The *average*, on the other hand, might be polluted by transient system interference (e.g., synchronization artifacts, GPU thermal throttling events, OS-induced delays, ...) unrelated to algorithmic efficiency.

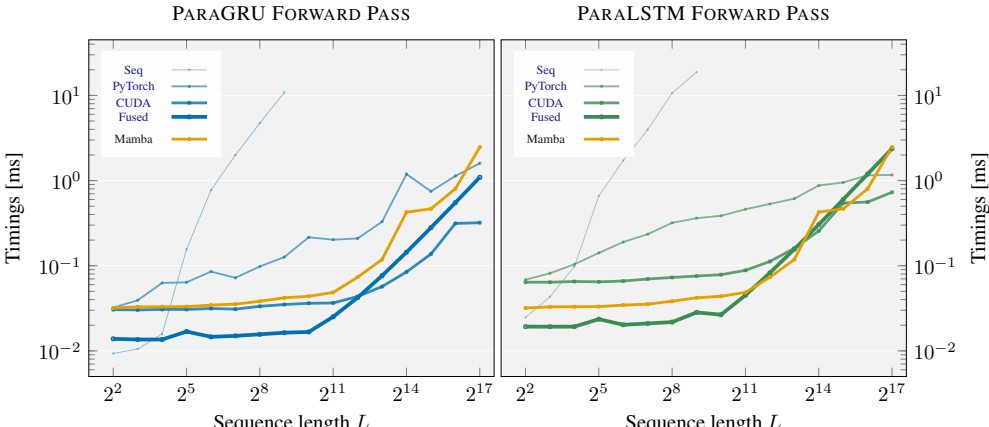

Figure 5: Timings of different application modalities for the ParaGRU (left) and ParaLSTM (right) RNN cells, applied to sequences of varying length. Heavier lines refer to progressively more efficient applications: sequential, using parallel reduction in PyTorch (Alg. 1a), using parallel reduction in CUDA and system assembly in PyTorch, and using a fully-fused CUDA kernel for the whole Newton routine. Equivalent timing results for Mamba are also included for reference (yellow). Same setup as for Fig. 2 (right).

perform (Jacobian assembly and Newton iterations, on top of parallel reduction). To ensure that a GPU block has enough resources to carry these computations, then, we are forced to reduce the chunk_size hyperparameter which controls the number of equations a single thread processes. This in turn triggers the block-sequential solution (hence the collapse to linear regime) for shorter sequence lengths than for the CUDA implementation ($L > 2^{10}$ instead of $L > 2^{11}$), thus explaining the behavior of the curves in Fig. 5 (left).

In Fig. 6 we further provide timings for a full parallel forward and backward application of our RNN cells, using a dummy loss function $\mathcal{L}(\boldsymbol{h}) = \sum_l \boldsymbol{h}_l^2$. The setup is analogous to that for Fig. 5, and the observations regarding relative performance of the various implementations remain substantially unchanged. Comparing with Fig. 5, however, we can notice that including the backward pass affects the timings for Mamba much more than that for ParaGRU or ParaLSTM. This is in line with expectations: in Mamba, both forward and backward passes involve the application of a *single* parallel reduction operation Alg. 1b. On the other hand, for nonlinear RNNs like ParaGRU or

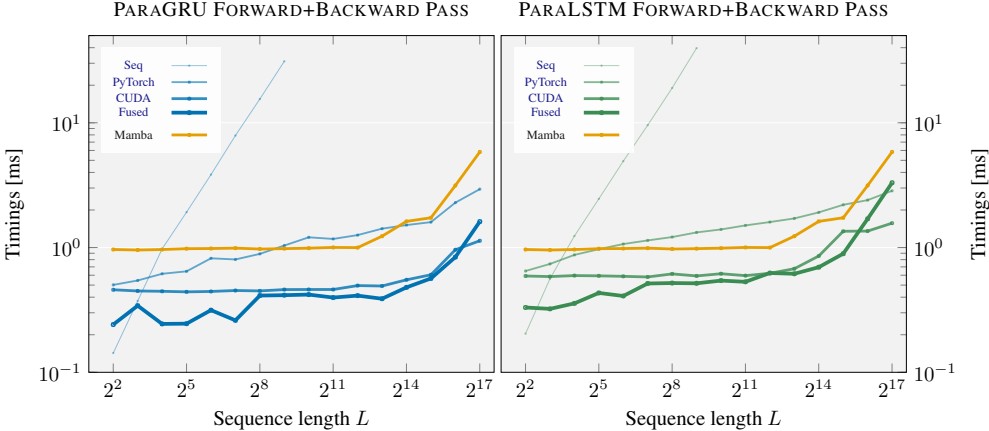

Figure 6: Timings of different application modalities for a full forward and backward pass of the ParaGRU (left) and ParaLSTM (right) RNN cells, applied to sequences of varying length. Same setup as for Fig. 5.

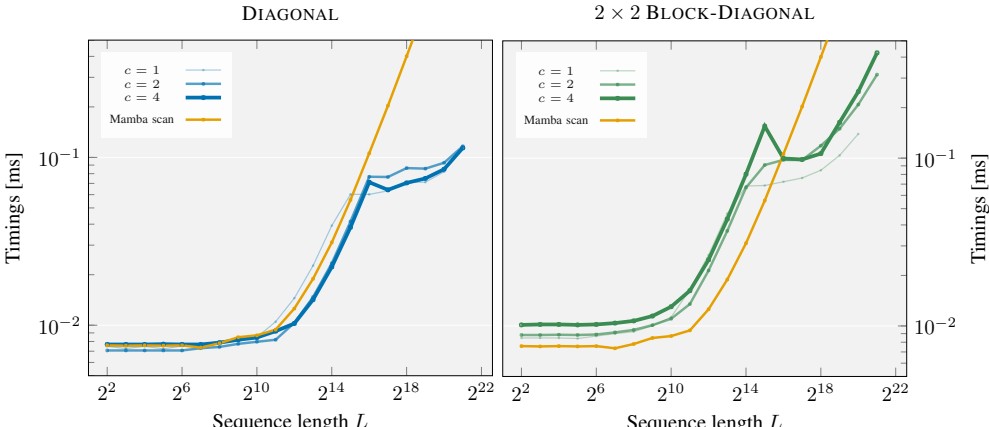

Figure 7: Timings of the CUDA implementation of parallel reduction, for diagonal (left) and $2 \times 2$ block-diagonal (right) Jacobians, varying its `chunk_size` hyperparameter $c$. Timings for Mamba implementation of `parallel_scan` are also reported as reference.

ParaLSTM, the forward pass requires *multiple* parallel reductions—one per each Newton iteration, that is $N_{\text{its}} = 3$ in our case—while the backward pass only requires a *single* parallel reduction, as outlined in Sec. 2.

**Ablation Over Kernel `chunk_size` Hyperparameter**    In App. D.2.2 we comment on the various compile-time hyperparameters one can tune to achieve the highest performance out of ParaRNN for their target setup. Likely the most relevant one is `chunk_size`, which defines the amount of work per thread—that is, the number of equations each thread must solve individually. Here we report the results of the sweep we conducted with the goal of selecting its optimal value. In particular, in Fig. 7 we show timings for the sole parallel reduction operation applied to diagonal (left, used by ParaGRU) and $2 \times 2$ block-diagonal Jacobians (right, used by ParaLSTM), when varying `chunk_size`. For the diagonal case, we pick a value of $c = 2$ as the optimal, as it achieves the best performance for $L \lesssim 2^{12}$, but $c = 4$ would perform marginally better for $L \gtrsim 2^{12}$. Also for the block-diagonal counterpart, we consider $c = 2$, as it performs the best in the regime relevant to our training $2^{10} \lesssim L \lesssim 2^{14}$, although for shorter sentences $c = 1$ might be preferable.

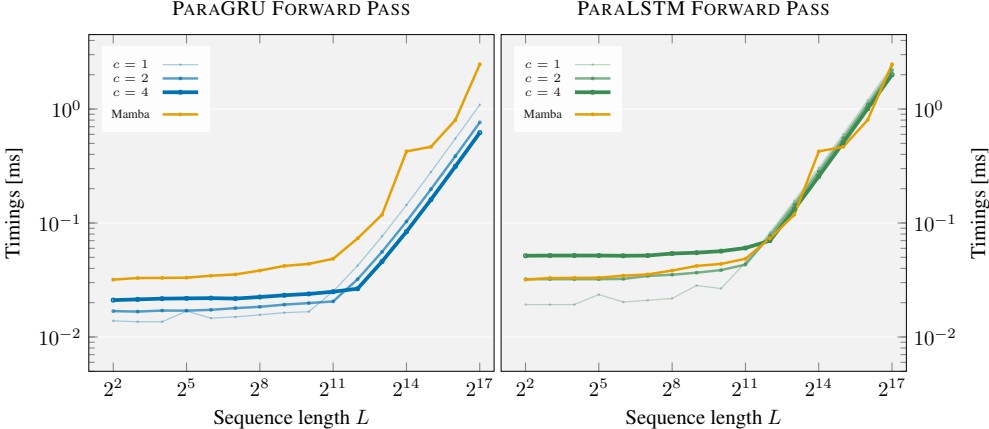

Figure 8: Timings of the implementation of the fully-fused kernel for parallel RNN application, for ParaGRU (left) and ParaLSTM (right), varying its `chunk_size` hyperparameter $c$. Timings for Mamba SSM application are also reported as reference.

We conduct a similar sweep also for the fully-fused parallel application of ParaGRU and ParaLSTM, and report its results in Fig. 8. For ParaGRU, the main advantage in increasing $c > 1$ consists in delaying the triggering of the linear regime, and indeed the choice of $c = 2$ is optimal for the $L = 2^{11}$ used in our training; for shorter sequences, however, sticking to $c = 1$ gives better runtimes. For ParaLSTM, this advantage is instead negligible, and using $c = 1$ throughout gives the best runtimes overall.

## B.2 ADDITIONAL THROUGHPUT RESULTS

In this section we expand on the results in Fig. 3, by reporting in Tab. 3 how throughput measurements vary with the scale of the models considered. The setup follows the same used for Fig. 3.

Overall, the trend is confirmed: the constant token generation cost of recurrent models allows them to reach much higher throughputs than Transformers, even for reasonably short sequence lengths (we report results for $L = 2^{11}$), and our RNNs perform marginally better than Mamba at all scales. Notice moreover that, for the recurrent cells considered, runtime is barely affected by the hidden dimensionality $d_h$. This is in line with the cell application being perfectly parallelizable across hidden state components, as granted by their (block-)diagonal structure. Indeed, when increasing only the hidden state size (which we do when moving from 400M to 1B models, and from 3B to 7B models—see also Tab. 4 for an outline of the model parameters considered at the various scales), throughput is only marginally impacted, mostly due to the presence of the additional MLP components (whose application runtime instead *does* scale with $d_h$). Conversely, throughput is mainly affected by increasing the model depth (passing from 1B to 3B), in line with expectations.

Table 3: Throughput results at regime ($L = 2^{11}$) for the model types and sizes considered.

| Model | 400M | 1B | 3B | 7B |
|---|---|---|---|---|
| ParaGRU | 38.12 | 38.09 | 28.58 | 28.45 |
| ParaLSTM | 36.77 | 36.93 | 27.06 | 27.68 |
| Mamba2 | 29.76 | 27.61 | 19.33 | 18.16 |
| Transformer | 6.45 | 1.98 | 1.00 | 0.42 |

## B.3 MEMORY PROFILING

In our work we consider the simplification to the Jacobian structure outlined in Sec. 3, so that *by design* the cell application requires storage scaling linearly in both sequence length $L$ and hidden state dimension $d_h$. In this section, we verify that this indeed holds in practice. To this end, in Fig. 9 we report how peak GPU memory usage recorded during the application of the ParaGRU and ParaLSTM cells varies when scaling $L$ or $d_h$. In both cases, perfect linear scaling is measured, with ParaLSTM requiring proportionally more storage, in light of its more complex Jacobians.

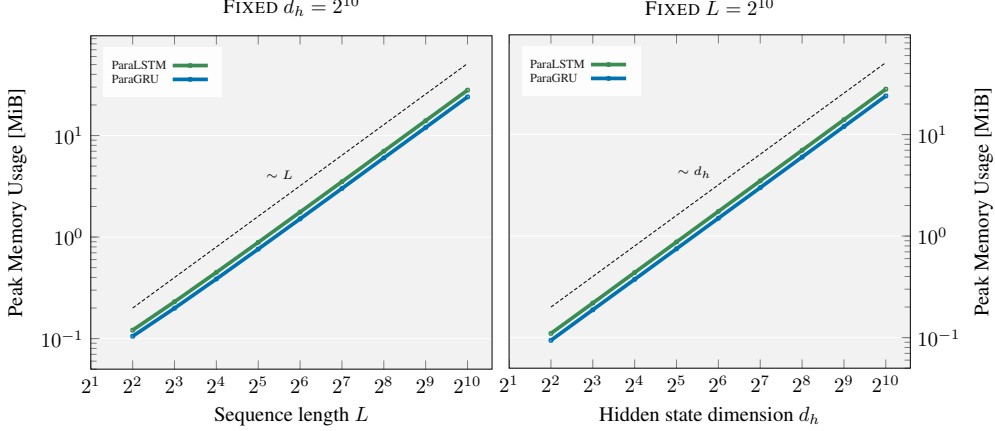

Figure 9: Peak GPU memory usage upon parallel application of the ParaGRU and ParaLSTM cells, varying sequence length $L$ (left) and hidden state dimension $d_h$ (right).

## C  DETAILS ON LANGUAGE MODELING RESULTS

**Summary**

In this section we provide more information about the architectures we used, the training parameters, timing protocol, and results regarding the language modeling experiments. In particular, in Tab. 6 we expand upon the results illustrated in Tab. 2, providing the downstream tasks evaluation scores across all model scales considered (400M, 1B, 2.9B, and 7B). Performance remains consistent across different parameter counts: with Mamba2, ParaGRU and ParaLSTM often outperforming the Transformer. Importantly, our RNN cells show consistent improvement with scale, and the relative performance gaps between architectures remain stable.

### C.1  LANGUAGE MODELING

**Models Description**  Schematics of the complete RNN block used for the language modeling experiments in Sec. 5 are shown in Fig. 10. The module inherits some components from Mamba (a prepending short sequence-wise causal convolution, as well as a gated RMS normalization layer following the actual RNN cell application), and includes a residual connection with a learnable scaling of the input. Still, our block follows closely the Transformer structure: particularly, MLP layers with residual connections are interwoven with the sequence mixer, which in our case is an RNN cell, rather than Attention. The core RNN cell is either ParaGRU or ParaLSTM from Sec. 3, but all other architectural components remain identical between the two variants. To aid with scaling, we use a multi-head implementation of our RNN cells, where the input $x_l \in \mathbb{R}^{d_{\text{in}}}$ is separated into $n_{\text{heads}}$ independent heads, each of size $d_h = d_{\text{in}}/n_{\text{heads}}$.

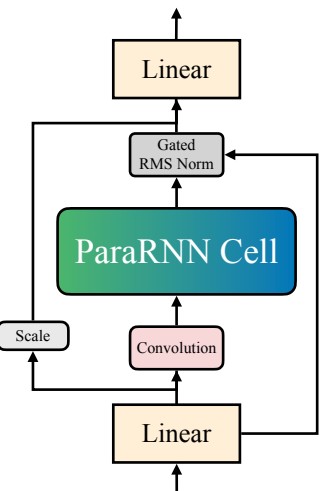

Figure 10: Schematics of RNN block.

The whole block is then repeated $n_{\text{layers}}$ times to generate the full model.

The baselines considered in our experiments are the Transformer architecture in DCLM (Li et al., 2024), and the Mamba2 model from HuggingFace. See Tab. 4 for a summary of hyperparameters further detailing the architectures used.

**Training Setup**  For our experiments we consider four architecture types—the ParaGRU and ParaLSTM introduced in this work, plus Mamba2 and Transformer as baselines: see Tab. 4 for architecture specifics—each at four different scales, of 400M, 1B, 2.9B and 7B parameters. As training dataset, we use SlimPajama (SPJ) (Soboleva et al., 2023), from which we removed the Books3 split (as it is known to contain copyrighted material). All models are trained using AdamW with detached weight decay and a cosine learning rate schedule, incorporating a warm-up for 10% of the total training iterations and decaying to a final learning rate of 0 (Bergsma et al., 2025). Training is conducted on nodes with 8 NVIDIA H100 GPUs each using PyTorch 2.6 with FSDP and native automatic mixed precision, storing weights in `bfloat16` while maintaining gradients and reduction operations in `float32`. Batch sizes and total number of training tokens follow Chinchilla scaling laws (Hoffmann et al., 2022), and we refer to Tab. 5 for the complete training configuration. Hyperparameter selection is conducted by sweeping over three learning rate and weight decay values for each model and each size, selecting the configuration resulting in lowest perplexity on the SPJ test set (see Tab. 4 for the optimal values selected in our runs). Model initialization follows architecture-

Table 4: Architecture specifications and optimal hyperparameters for our experiments. We report structural parameters (number of layers, number of heads and model width) and training hyperparameters (learning rate and weight decay) that achieved lowest perplexity for each model type and scale. Time measurements show average single-step training cost on NVIDIA H100 GPUs, while Memory measurements show the per-sample peak memory usage during one optimization step.

| | Model type | #params | $n_{\text{layers}}$ | $n_{\text{heads}}$ | $d_{\text{model}}$ | Learning rate | Weight decay | ↓ Time [ms/step] | ↓ Memory [GiB/sample] |
|---|---|---|---|---|---|---|---|---|---|
| 400M | ParaGRU | 406M | 24 | 4 | 1024 | 0.005 | 1e-3 | 95.7 | 6.2 |
| | ParaLSTM | 406M | 24 | 4 | 1024 | 0.003 | 1e-3 | 116 | 6.2 |
| | Mamba2 | 409M | 48 | 32 | 1024 | 0.005 | 1e-3 | 115 | 5.2 |
| | Transformer | 412M | 24 | 8 | 1024 | 0.003 | 1e-3 | 81.0 | 5.3 |
| 1B | ParaGRU | 1.42B | 24 | 4 | 2048 | 0.003 | 1e-4 | 192 | 9.8 |
| | ParaLSTM | 1.42B | 24 | 4 | 2048 | 0.003 | 1e-4 | 235 | 9.8 |
| | Mamba2 | 1.43B | 48 | 64 | 2048 | 0.005 | 1e-4 | 221 | 9.1 |
| | Transformer | 1.44B | 24 | 16 | 2048 | 0.002 | 1e-4 | 157 | 8.5 |
| 2.9B | ParaGRU | 2.90B | 32 | 2 | 2560 | 0.002 | 1e-4 | 326 | 16.8 |
| | ParaLSTM | 2.90B | 32 | 2 | 2560 | 0.002 | 1e-4 | 400 | 16.9 |
| | Mamba2 | 2.83B | 64 | 80 | 2560 | 0.003 | 1e-4 | 393 | 17.2 |
| | Transformer | 2.80B | 32 | 32 | 2560 | 0.003 | 1e-4 | 267 | 14.1 |
| 7B | ParaGRU | 6.76B | 32 | 4 | 4096 | 0.003 | 1e-4 | 613 | 38.2 |
| | ParaLSTM | 6.76B | 32 | 4 | 4096 | 0.003 | 1e-4 | 715 | 39.3 |
| | Mamba2 | 6.96B | 64 | 128 | 4096 | 0.003 | 1e-4 | 9475 | 39.8 |
| | Transformer | 6.89B | 32 | 32 | 4096 | 0.003 | 1e-4 | 520 | 31.0 |

specific strategies: for Mamba2, we use the default initialization available in HuggingFace. For Transformers, we use DCLM initialization (Li et al., 2024). Since our ParaGRU and ParaLSTM blocks in Fig. 10 mimic the Transformer architecture, we opt for DCLM initialization for them as well, except for their core RNN cells, whose initialization depends on the parameter considered: for the cell input matrices $B_\star$ in (7) we use *Kaiming Uniform* (He et al., 2015) initialization; for the (diagonal) state and peephole matrices $a_\star$ and $c_\star$ in (9) we choose *Xavier Gaussian*, since we found small values improved stability; the bias terms $b_\star$ are instead initialized to $\mathbf{0}$. We also clip the norm of $a_\star$ and $c_\star$ to a maximum of $0.5$ to prevent overflows in the computation of the hidden state for long sequences: this effectively bounds the Jacobians in (8), similarly to what is done in Mamba, where the state matrix eigenvalues are constrained in $(0, 1)$.

To illustrate the stability of our training procedure, Fig. 11 shows training and validation loss curves for our 7B-parameter models. Notably, both ParaGRU and ParaLSTM exhibit smooth convergence throughout training—especially when compared to the baselines—suggesting that the gradient flow in our recurrent models is well-behaved at scale.

**Additional Results** In Tab. 6 we expand upon the results illustrated in Tab. 2, providing the downstream tasks evaluation scores across all model scales considered (400M, 1B, 2.9B, and 7B). Over-

Table 5: Scale-specific training configurations on nodes with 8 NVIDIA H100 GPUs each All models use 2048-token sequences and follow ($\sim 1\times$) Chinchilla-optimal token budgets (Hoffmann et al., 2022) for batch-size selection. The z-loss regularization coefficient is taken from DCLM (Li et al., 2024).

| Model Scale | Nodes | Batch Size per GPU | Effective Batch Size | Z-Loss | Total Iterations | Warm-up Iterations | Training Tokens [B] |
|---|---|---|---|---|---|---|---|
| 400M | 1 | 32 | 256 | 1e-04 | 20,000 | 2,000 | 10.5 |
| 1B | 1 | 32 | 256 | 1e-04 | 55,000 | 5,500 | 28.8 |
| 2.9B | 4 | 16 | 512 | 1e-04 | 55,000 | 5,500 | 57.7 |
| 7B | 32 | 8 | 2,048 | 5e-06 | 33,000 | 3,300 | 138.4 |

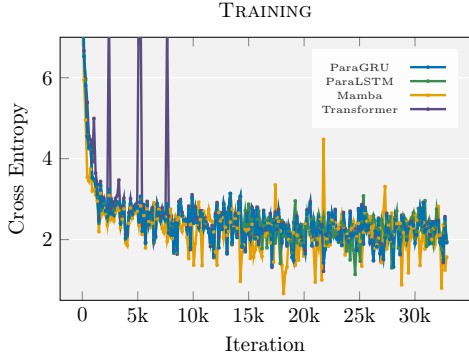 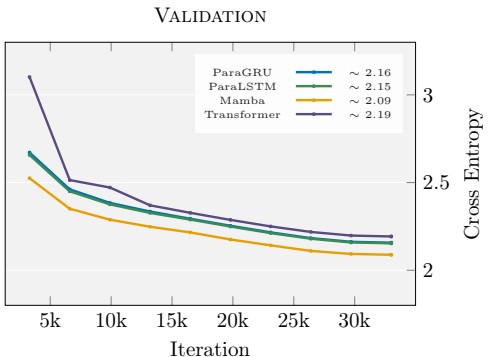

Figure 11: Cross-entropy loss evolution for the 7B models considered, evaluated on the training (left, collected every 150 iterations), and validation (right, collected every 3.3k iterations) datasets.

all, the performance observed is consistent across different parameter counts: while Mamba2 still achieves the lowest perplexity, ParaGRU and ParaLSTM remain competitive, with all three recurrent architectures often outperforming the Transformer. Importantly, our RNN cells show consistent improvement with scale, and the relative performance gaps between architectures remain stable.

## C.2 SYNTHETIC TASKS

To recover the results in Tab. 1, we trained a *single layer* of ParaGRU, ParaLSTM, Mamba2, and Transformer across a variety of synthetic tasks that have been used in the literature to evaluate the expressivity of sequential models. The evaluation tasks considered are:

**Keep-n$^{th}$** Given an input sequence $[x_l]_{l=1}^L$ of values from a dictionary $x_l \in V$, output its $n$-th element, $x_n$.

**Parity** Given an input sequence of values in $\{0, 1\}$, output $\mod (\sum_{l=1}^L x_l, 2)$.

**Multiple-Query Associative Recall (MQAR)** Given an input sequence of key-value pairs $(k, v) \in K \times V$, followed by noise interwoven with the keys, output for each key its associated value.

Table 6: Final perplexities, and evaluation scores on reference downstream tasks from `lm-eval-harness` (Gao et al., 2021), for all model types and scales considered.

|  | Model | #params | ↓ PPL | ↑ Arc-C (25) | (3) | ↑ HSwag (10) | (0) | ↑ OBQA (10) | (0) | ↑ WinoG (5) | (0) | ↑ PiQA (0) | ↑ MMLU (0) |
|---|---|---|---|---|---|---|---|---|---|---|---|---|---|
| 400M | Transformer | 412M | 17.00 | 24.23 | 23.81 | 36.11 | 35.91 | 26.60 | 29.40 | 51.77 | 51.69 | 64.64 | 25.37 |
|  | ParaGRU | 406M | 18.56 | 22.27 | 21.93 | 38.69 | 39.21 | 25.00 | 25.80 | 53.04 | 52.41 | 67.30 | 23.68 |
|  | ParaLSTM | 406M | 18.40 | 23.12 | 24.23 | 37.43 | 37.52 | 26.20 | 26.20 | 51.22 | 49.49 | 65.94 | 23.37 |
|  | Mamba2 | 409M | 16.49 | 24.32 | 25.09 | 40.53 | 41.00 | 30.00 | 34.00 | 50.99 | 51.14 | 66.97 | 23.41 |
| 1B | Transformer | 1.44B | 12.90 | 26.87 | 25.51 | 47.25 | 47.14 | 34.00 | 33.40 | 54.06 | 54.22 | 68.49 | 23.12 |
|  | ParaGRU | 1.42B | 13.20 | 24.15 | 23.81 | 48.71 | 48.85 | 24.80 | 25.80 | 53.43 | 53.28 | 71.27 | 24.55 |
|  | ParaLSTM | 1.42B | 13.11 | 24.15 | 24.32 | 45.82 | 46.53 | 24.40 | 27.00 | 51.14 | 51.07 | 70.46 | 24.75 |
|  | Mamba2 | 1.43B | 12.26 | 28.41 | 27.47 | 50.90 | 51.21 | 36.20 | 36.00 | 55.25 | 55.17 | 71.71 | 24.39 |
| 2.9B | Transformer | 2.80B | 10.97 | 30.72 | 29.52 | 55.59 | 55.20 | 36.80 | 35.40 | 57.06 | 58.41 | 73.12 | 23.12 |
|  | ParaGRU | 2.90B | 10.92 | 23.98 | 23.12 | 57.58 | 57.94 | 26.00 | 26.60 | 53.83 | 56.51 | 74.48 | 23.75 |
|  | ParaLSTM | 2.90B | 10.87 | 24.15 | 24.06 | 54.22 | 55.14 | 26.80 | 27.40 | 53.83 | 53.75 | 71.71 | 23.99 |
|  | Mamba2 | 2.83B | 10.13 | 34.47 | 33.70 | 61.13 | 61.12 | 42.20 | 37.80 | 59.43 | 59.51 | 73.78 | 24.89 |
| 7B | Transformer | 6.89B | 9.55 | 34.30 | 33.36 | 62.98 | 62.20 | 40.00 | 37.20 | 61.48 | 60.85 | 74.97 | 23.12 |
|  | ParaGRU | 6.76B | 9.19 | 39.68 | 36.77 | 65.85 | 65.75 | 42.20 | 40.40 | 61.40 | 59.83 | 76.66 | 25.29 |
|  | ParaLSTM | 6.76B | 9.16 | 37.46 | 36.52 | 62.47 | 62.85 | 42.20 | 38.80 | 57.70 | 59.12 | 75.19 | 25.31 |
|  | Mamba2 | 6.96B | 8.62 | 40.02 | 39.59 | 69.78 | 69.68 | 42.20 | 42.20 | 65.19 | 63.77 | 76.66 | 26.61 |

$k$-**hops** Given an input sequence of values from a dictionary $V$, for each element in the sequence "hop back" and report the value following its previous occurrence in the sequence. Keep on hopping back $k$ times. The special case of 1-hop is also known as *induction heads*.

**Cycle Navigation** Given a sequence of steps $x_l \in \{-k, \ldots, k\}$ on a cycle of size $C$, output the final position $\mod (\sum_{l=1}^{L} x_l, C)$.

**Modular Arithmetic (w/o brackets)** Given a sequence of digits in $V$ and operators $\{+, -, *\}$ forming an arithmetic expression, output the result modulo $|V|$, respecting standard order of operations.

**Copy Memory** Given a sequence $[x_l]_{l=1}^{M}$ of values from a dictionary $V$, followed by blank tokens and a trigger at $x_L$, output the original sequence.

**A5** Given a sequence of elements from the alternating group $A_5$ (even permutations), output the cumulative group products at each position.

**Models Description** For all models we train a single layer. The models under test are structured as: embedding layer with size matching the task vocabulary size, RMSNorm normalization layer, attention / SSM / RNN block, another RMSNorm normalization layer, and a linear layer with size matching the task vocabulary size. For Keep-n$^{\text{th}}$ and Parity described below, the SSM and RNN blocks contain *only* the mixer (in the tables we highlight this with $^{\dagger}$), that is we remove the convolutional layer, normalization layer, and gating mechanism from the block definition (see Fig. 10). We do this to test the expressivity of the mixer themselves, which is the main difference between the models. For the remaining tasks we use instead the full block (as described in App. C.1 and Fig. 10) since the performance of both Mamba and the RNNs dropped significantly without these additional components. Moreover, for Keep-n$^{\text{th}}$ we add positional encoding to Mamba and the RNN cells since it is not possible to solve the task without it (the Transformer includes positional encoding in all tasks). Across all models, we set a model dimension of 64, split across 4 heads. The convolutional layers (when used) have a kernel size of 4.

**Training Setup** For MQAR, $k$-hop, Keep-n$^{\text{th}}$ and Parity, we randomly obtain 10k samples as the training set and 100k samples as the test set (which we report on), with sequence length fixed to $L = 100$. We train each architecture for up to 3,000 epochs with a batch size of 16. For the remaining tasks, the training set contains 1M samples and the test set 100k samples, with sequence length varying by task: $L = 100$ for Cycle Navigation, $L = 50$ for Copy Memory, and $L = 20$ for A5 and Modular Arithmetic. We train these tasks for up to 1,000 epochs with a batch size of 512. For optimization, we employ an AdamW optimizer ($\beta_1 = 0.9$, $\beta_2 = 0.999$), cosine-scheduled learning rate starting at $5 \times 10^{-4}$, and weight decay of $10^{-6}$. All tasks are trained on a single NVIDIA A100 GPU, across three different seeds (we report on the best result). The norm of $\boldsymbol{a}_\star$ and $\boldsymbol{c}_\star$ in ParaGRU and ParaLSTM is clipped to a maximum of 0.90, except for Parity, where we found that norm clipping was detrimental to the cells' ability to solve the task.

## D  PARARNN SOFTWARE PACKAGE OVERVIEW

In this section we provide an overview of the ParaRNN software package developed in the course of this research work. In its design we focused on three principles: modularity, ease of use, and extensibility. Consequently, the code separates the definition of the RNN cell and the solver, and drastically reduces the amount of code needed to be implemented by practitioners. Additionally, we provide multiple hooks for extension and generalization of various RNN cells.

### D.1  MAIN CLASSES

The core interface of the package revolves around three main classes, respectively responsible for providing abstractions to an RNN Cell, the implementation of its action, and the desired method for its application. More in detail:

**RNN Cell**   The `BaseRNNCell` class represents the abstraction of an actual RNN cell, implemented as a torch module: all user-defined cells should inherit from it. This is the only stateful class in the package, and as such it is responsible for initializing and storing the parameters defining the cell itself. These are collected in a single `SystemParameters` dataclass, to aid interoperability with the other classes in ParaRNN by providing a standardized interface for parameter access. The actual implementation of the key RNN cell functions *is kept separate* from the class definition itself, and delegated to the static class `RNNCellImpl`. `BaseRNNCell` is also responsible for coordinating code execution, invoking the correct method from `RNNCellApplication` depending on the selected RNN cell application modality.

**RNN Cell Implementation**   The base static class `RNNCellImpl` collects the implementation of the functions necessary to describe the action of an RNN cell. Its main responsibility is to implement the reccurrence step (1). Starting from this, the class leverages PyTorch `autograd` functionalities to automatically define the necessary methods to assemble the target all-at-once system (2), and particularly the computation of residuals and Jacobians in (3). For additional performance, it is possible to override the latter function, and provide an explicit definition of the Jacobians—we do this ourselves in the implementation of the ParaGRU and ParaLSTM cells. This becomes particularly relevant when said Jacobians present a specific structure, that can be leveraged for making the reduction operations involved in the solution of (3) more computationally tractable. To this aim, in ParaRNN we provide two specializations for `RNNCellImpl`: `RNNCellDiagImpl` and `RNNCellBlockDiagImpl`, which simplify operations for those cells whose Jacobians are respectively diagonal (like for ParaGRU), or composed of $N \times N$ diagonal blocks (like for ParaL-STM, where $N = 2$). These specializations implement the corresponding reduction steps (lines 4 and 5 in Alg. 1b), described more in detail in App. D.2. Our software allows to further specialize `RNNCellImpl` to encompass additional Jacobians structures (e.g., sparse, or Circulant (Keller et al., 2024)), but the corresponding assembly, storage, and reduction operations should be provided (see also App. D.4).

**RNN Cell Application**   The third main component of ParaRNN is `RNNCellApplication`. This collection of static classes defines the different ways in which the RNN cell can be applied, and effectively implements the forward and backward passes for `BaseRNNCell`. The most vanilla version, `RNNCellSequentialApplication`, simply evaluates the cell output by sequentially applying (1) to the input sequence: this is the classical way PyTorch RNN cells act, although in our work this is mostly used for testing and debugging, or at inference time. `RNNCellParallelApplication` implements the sequence-parallel application of the RNN cell, following the method described in Sec. 2, while using only PyTorch directives: the class effectively acts as a wrapper for our implementation of Newton's method, and leverages the methods from `RNNCellImpl` to iteratively assemble and solve the linearized system (3) via parallel reduction. A first optimization is provided in `RNNCellParallelCUDAApplication`, where the parallel solve is handled by a custom CUDA kernel, while the system assembly is still performed in PyTorch. Finally, `RNNCellParallelFusedApplication` bypasses all PyTorch-based implementations and computes the RNN application via a fully-fused CUDA kernel (implementing Newton's iterations, system assembly, and parallel reduction, end-to-end).

### D.2    Solver implementations

In addition to a programming interface for RNN cells, the main contribution of ParaRNN consists in providing ready-to-use efficient implementations of the parallel solver outlined in Sec. 2.

#### D.2.1    Pure PyTorch parallel reduction

Our PyTorch reference implementation is a direct adaptation of the pseudocode in Alg. 1a and 1b. Particularly, the parallel reduction loop reduces to

```
@staticmethod
def parallel_reduce(
        jac: torch.Tensor,     # (B),L,?,? -> shapes will depend on
        rhs: torch.Tensor,     # (B),L,?       specific jac structure
        reduction_step: typ.Callable
) -> torch.Tensor:
    num_steps = math.ceil(math.log2(rhs.shape[-2]))
    for step in range(num_steps):
        jac, rhs = reduction_step(jac, rhs, step)
    return rhs
```

The `reduction_step` method defines how to perform pairwise reduction of the equations in (3): namely, it implements lines 4 and 5 of Alg. 1b, adapted to a specific structure of the cell Jacobians. This is the only cell-specific method used in the solver, and by treating it as an abstraction, we are allowing for modularity in its utilization. In particular, in ParaRNN we provide implementations of this method for dense, diagonal, and block-diagonal Jacobians, covering both the general and the specific cases analyzed in this paper (c.f. Sec. 3).

The main focus of the development is dedicated to the design and implementation of the CUDA-accelerated solver. We describe this in detail in the following.

#### D.2.2    CUDA-accelerated parallel reduction

Each level of GPU execution has access to progressively slower but larger memory systems (NVIDIA, 2025, Sec. 5.2-5.3). *Threads* can access their own registers the fastest, *warps* can shuffle data between threads seamlessly, and *blocks* can use dedicated shared memory paying a small overhead. Only when communication occurs at *grid* level must we rely on the slow global memory of the device. In implementing our kernel for parallel reduction, we minimize overhead by closely mirroring this hierarchy: the resulting algorithm directly adapts the reduction operations to each level. On top of this, rather than blindly maximizing parallelism at every level of the hierarchy, we follow a hybrid approach interweaving both sequential and parallel reductions, inspired by similar work on solvers for tri-diagonal systems (László et al., 2016). The action of our algorithm as it traverses the GPU hierarchy is outlined next.

**Thread-level**    At the lowest hierarchy level in our algorithm, we associate each GPU thread with a small group of `chunk_size` adjacent equations in the target system (3). Threads are responsible for reading from global memory to their registers the information (namely Jacobians and residuals) associated with their chunk of equations. At this stage in the algorithm, threads must also manipulate said information and start reducing equations so that they all end up depending solely on the value of the very first unknown in their chunk. This is obtained by *forward substitution*, i.e., by sequentially reducing each equation in a chunk using the previous one. Note that, while this operation is sequential, it only requires `chunk_size` $\sim 2 - 4$ steps, and is nonetheless conducted in parallel across threads. See Fig. 12 for a diagram of the action of the algorithm at this level.

**Warp-level**    The first truly parallel reduction operations start occurring at warp level. Threads within a same warp share information from the last equations in their chunk and apply parallel reduction over them. At the end of this operation, each of these equations depends solely on the very first one in the warp. Explicitly leveraging this level in the hierarchy of the algorithm enables the use of warp-specific directives (such as warp shuffle instructions) for threads to efficiently access information stored in other threads registers, eliminating the need for expensive memory transfers or explicit synchronization. See Fig. 13 for a diagram of the action of the algorithm at this level.

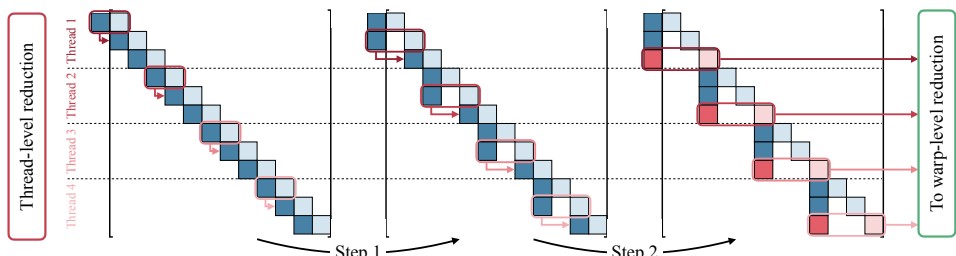

Figure 12: Schematics drafting the evolution of the target block bi-diagonal system (3) as it undergoes the first stage of our algorithm. Light-blue squares represent its main diagonal (i.e., identity matrices), while the dark-blue once represent the Jacobians at different $l$'s. Threads independently apply forward substitution to chunk_size($= 3$ in the figure) equations. At each step, this amounts to using equation $l$ to reduce the following one $l + 1$ (i.e., applying lines 4 and 5 in Alg. 1b). Notice this makes all equations in a chunk depend on the value of the very first unknown, as exemplified by the progressive shift towards the left of the sub-diagonal block. At the end of this stage, we can effectively extract a simplified system involving only the last equation of each chunk, which is then passed onto the next stage.

**Block-level** Parallel reduction proceeds then in a similar fashion at block level, where the last equations of each warp are reduced so that they depend only on the very first one in the block. For this stage, we need to resort to the shared memory within the block to share information among warps, access to which must be explicitly synchronized: while this incurs some overhead, it remains significantly faster than accessing the device global memory. Once this stage is completed, we can traverse the hierarchy in reverse. Specifically, we use the first equation in the block to solve in parallel for the last equations of each warp, then use these to solve in parallel for the last equations of each chunk, and finally use these to solve in parallel for each equation within a chunk. If a single block is large enough to cover the whole input sequence, then at this stage the solution to (3) is recovered, and the algorithm terminates here, by writing the solution back to global memory. If not, the next stage depends on the total sequence length. If it is smaller than a certain threshold, then the same block sequentially processes consecutive segments of the sequence, restarting the procedure above. For very long sequences, instead, the next level in the hierarchy is triggered.

**Grid-level** For extremely long sequences, rather than having a single block covering sequentially the whole sequence length (while still applying parallel reduction within the block), we organize the GPU kernel grid so that multiple blocks are fired to cover the sequence in parallel, applying the

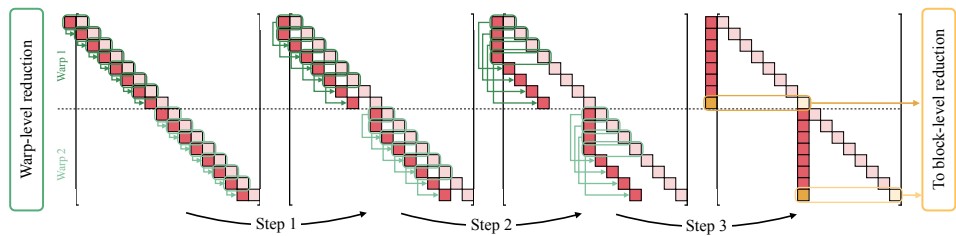

Figure 13: Schematics drafting the evolution of the simplified block bi-diagonal system stemming from thread-level reduction, as it undergoes the second stage of our algorithm. The threads within a same warp (8 in the figure, but 32 in practice) apply parallel reduction to further simplify the target system. At each step $i$, this amounts to using equation $l$ to reduce the $l + 2^i$-th one. Notice this progressively shifts the sub-diagonal blocks by $2^0, 2^1, 2^2, \ldots$ positions, until all equations depend only on the very first unknown in the warp. At the end of this stage, we can effectively extract a simplified system involving only the last equation of each warp, which is then passed onto the next stage. At the block level, the algorithm proceeds similarly: the last threads of each warp perform parallel reduction, operating on the simplified system.

same procedure listed above. In this case, however, the procedure leaves each block dependent on the value of its first unknown (whose solution is explicitly available only for the very first block, which covers the start of the sequence). An additional parallel reduction is then necessary to solve for the last equations of each block. This requires waiting for the completion of the kernel performing reduction *within* a block, firing a second kernel to reduce equations *across* blocks, and then a third one to perform the final substitutions within each block again once their solutions are available. The associated overhead grows accordingly, not only due to the additional kernel launches, but also because in-between kernels execution the device must be synchronized, and partial results must be written to global memory.

The size of the chunk of equations solved by each thread (chunk_size), the number of threads allocated to a single block (threads_per_block), and the threshold number of sequential block-wise reductions (max_sequential_steps) are all hyperparameters defined at compile-time, which can be tuned to squeeze additional performance from the code by changing the per-thread workload and registers availability. The most optimal combination of parameters is a function of the actual GPU device in use, the underlying scalar datatype considered, the target input sequence length, and of the Jacobian structure of the RNN cell. In Fig. 8 we report timing results for a sweep on chunk_size, which we used to select the most optimal values in this work. The defaults provided in the code are a good starting point for a typical LLM training setup in mind, (i.e., sequences of $L = 2^{9 \sim 12}$, float32 scalars), on a NVIDIA A100 GPU.

## D.3 EXAMPLE USE

In the following, we provide an example implementation of a simplified version of the ParaGRU cell used in this work, as a use-case for the ParaRNN codebase.

As a first step, we need to introduce the parameters involved in the cell definition itself:

```python
@dataclass
class GRUDiagSystemParameters(SystemParameters):
    A: torch.Tensor      # combines (Az,Ar,Ac)
    B: torch.Tensor      # combines (Bz,Br,Bc)
    b: torch.Tensor      # combines (bz,br,bc)
    nonlin_update: typ.Callable
    nonlin_reset: typ.Callable
    nonlin_state: typ.Callable
```

These will be stored and initialized inside our new cell:

```python
class GRUDiag(BaseRNNCell[
    GRUDiagConfig, GRUDiagSystemParameters, GRUDiagImpl
]):
    # invoked by base constructor
    def _specific_init(self, config: GRUDiagConfig):
        def __init__(self, config: GRUDiagConfig):
            super().__init__(config)

        # System parameters
        # - collated parameters for computing z,r,c
        kwargs = {"device": self.device, "dtype": self.dtype}
        self.A = nn.Parameter(torch.empty([3, self.state_dim], **kwargs))
        self.B = nn.Parameter(torch.empty(
            [3, self.state_dim, self.input_dim],**kwargs)
        )
        self.b = nn.Parameter(torch.empty([3, self.state_dim], **kwargs))
        self.nonlin_update = self._set_nonlinearity(config.nonlin_update)
        self.nonlin_reset = self._set_nonlinearity(config.nonlin_reset)
        self.nonlin_state = self._set_nonlinearity(config.nonlin_state)

        # Parameter initialisation
        self.reset_parameters()

    @property
    def _system_parameters(self):
        # - handily collect system parameters
        return GRUDiagSystemParameters(
            A=self.A, B=self.B, b=self.b,
            nonlin_update=self.nonlin_update,
            nonlin_reset=self.nonlin_reset,
```

```
            nonlin_state=self.nonlin_state,
        )
```

The final ingredient to be able to start using the class, is prescribing its action (7a):

```python
class GRUDiagImpl(RNNCellDiagImpl[GRUDiagSystemParameters]):
    @staticmethod
    def recurrence_step(
            x: torch.Tensor,      # (B), (L), Din
            h: torch.Tensor,      # (B), (L), Dh
            system_parameters: GRUDiagSystemParameters
    ) -> torch.Tensor:
        Bxpb = torch.einsum(
            '...j,vij->...vi',
            (x, system_parameters.B)
        ) + system_parameters.b
        AhpBxpb = torch.einsum(
            'vj,...j->...vj',
            (system_parameters.A[:2, :], h)
        ) + Bxpb[..., :2, :],
        z, r = torch.unbind(AhpBxpb,dim=-2)
        z = system_parameters.nonlin_update(z)
        r = system_parameters.nonlin_reset(r)
        c = system_parameters.nonlin_state(
            system_parameters.A[2, :] * h * r + Bxpb[..., 2, :]
        )
        return (1 - z) * h + z * c
```

This is the minimum implementation the user must provide to be able to apply the RNN cell. Just with this, the user can already toggle between sequential and parallel application[3] by selecting `model.mode=SEQUENTIAL` or `model.mode=PARALLEL`. Moreover, since the Para-GRU cell specified above presents a diagonal structure, the user can directly leverage the efficient implementations provided in `RNNCellDiagImpl` for Jacobians assembly (via `autograd`) and manipulation, and the corresponding CUDA-accelerated solver is also available via selecting `model.mode=PARALLEL_CUDA`. Still, by manually providing the formula for the Jacobians (8), one can boost performance by avoiding the reliance on `autograd`:

```python
class GRUDiagImpl(RNNCellDiagImpl[GRUDiagSystemParameters]):
    @classmethod
    def compute_jacobians(
            cls,
            h: torch.Tensor,
            x: torch.Tensor,
            system_parameters: GRUDiagMHSystemParameters,
    ) -> torch.Tensor:
        hm1 = cls._roll_state(h)          # shift by 1 along L
        Bxpb = torch.einsum(
            '...j,vij->...vi',
            (x, system_parameters.B)
        ) + system_parameters.b
        pre_nl_z, pre_nl_r = torch.unbind(
            torch.einsum(
                'vj,...j->...vj',
                (system_parameters.A[:2, :], hm1
            )) + Bxpb[..., :2, :],
            dim=-2
        )
        z = system_parameters.nonlin_update(pre_nl_z)
        r = system_parameters.nonlin_reset(pre_nl_r)
        pre_nl_c = system_parameters.A[2, :] * hm1 * r + Bxpb[..., 2, :]
        c = system_parameters.nonlin_state(pre_nl_c)

        grad_z = system_parameters.derivative_nonlin_update(pre_nl_z)
        grad_r = system_parameters.derivative_nonlin_reset(pre_nl_r)
        grad_c = system_parameters.derivative_nonlin_state(pre_nl_c)

        J_z, J_r, J_c = torch.unbind(
            system_parameters.A * torch.stack(
                [grad_z, grad_r, grad_c], dim=-2
```

---

[3]Notice that the `recurrence_step` method is designed to work regardless of whether the input is batched or not, and whether it is a full sequence or a single element: this allows to reuse the exact same method for the different application modes of the RNN cell.

```
            ), dim=-2
        )
        J_c = J_c * (r + hm1 * J_r)

        jac = (1 - z) + (h - hm1) * J_z + z * J_c

        return - jac
```

Finally, to be able to use the most optimized application mode `model.mode=PARALLEL_FUSED`, it suffices for the user to provide equivalent CUDA implementations for the functions above. In particular:

```cpp
template<typename scalar_t> // Implements CRTP for static polymorphism
class GRUCellDiagImpl : public RNNCellDiagImpl<
    scalar_t, GRUCellDiagImpl< scalar_t >
> {
public:
    __device__ static void readDataFromGlobal(
        const scalar_t*... glbVars,
        lclVars_t& lclVars
    ){
        // Read torch tensors in system_parameters (glbVars) and store
        //  thread-specific info in registers (lclVars)
        ...
    }

    __device__ static void recurrenceStep(
        const rhs_t  &h,
        const lclVars_t& lclVars,
        rhs_t  &hp1
    ){
        auto& [a, Bxpb] = lclVars;
        scalar_t z = nonlinUpd(    a[0] * h     + Bxpb[0] );
        scalar_t r = nonlinRes(    a[1] * h     + Bxpb[1] );
        scalar_t c = nonlinState( a[2] * h * r + Bxpb[2] );
        hp1 = (1.-z) * h + z * c;
        return;
    }

    __device__ static void computeJacobians(
        const rhs_t  &h,
        const rhs_t  &hm1,
        const lclVars_t& lclVars,
        jac_t  &jac
    ){
        auto& [a, Bxpb] = lclVars;
        scalar_t z = a[0] * hm1 + Bxpb[0];
        scalar_t r = a[1] * hm1 + Bxpb[1];
        scalar_t jz = a[0] * derivativeNonlinUpd(z);
        scalar_t jr = a[1] * derivativeNonlinRes(r);
        z = nonlinUpd(z);
        r = nonlinRes(r);
        scalar_t c =  a[2] * hm1 * r + Bxpb[2];
        scalar_t jc = a[2] * derivativeNonlinState(c) * ( r + hm1 * jr );
        c = nonlinState(c);
        jac = - ( (1. - z) + (c - hm1) * jz + z * jc);
        return;
    }
};
```

### D.4 EXTENSIONS TO OTHER JACOBIANS STRUCTURES

For RNNs with Jacobians structures different from (block-)diagonal, the user has two choices: either fall-back to the implementation for generic dense Jacobians (the base `RNNCellImpl`), or provide their own specialization. This in particular requires defining two core methods:

```python
class MyStructuredCellImpl(
    RNNCellImpl[SystemParametersT], typ.Generic[SystemParametersT]
):
    @classmethod
    def compute_jacobians(
            cls,
```

```
        h: torch.Tensor,
        x: torch.Tensor,
        system_parameters: SystemParametersT,
) -> torch.Tensor:
    ...

@classmethod
def parallel_solve(cls) -> typ.Callable:
    return partial(parallel_reduce, reduction_step=...)
```

The method `compute_jacobians` not only should take care of the actual assembly of the Jacobians, but also implicitly defines how they will be represented. This must be taken into account in the definition of the specific `reduction_step` function (which also must be implemented), responsible of describing how to conduct the reduction operations (see App. D.2.1 and lines 4 and 5 in Alg. 1b) for this specific Jacobian structure. For the largest part, the structure of our CUDA code follows a similar pointer-to-implementation design pattern, mirroring that of the PyTorch code. In principle, then, one can provide a similar specialization for the Jacobians structure also in CUDA:

```
template < typename scalar_t, typename Derived = void >
class MyStructuredCellImpl : public RNNCellBaseImpl<
    std::conditional_t< std::is_void_v< Derived >,
        MyStructuredCellImpl< scalar_t >,
        Derived
    >       // Implements CRTP for static polymorphism
> {
    // Reduce current eq using other
    __device__ static void reduceEqs(
        const jac_t& jacOther, jac_t& jac,
        const rhs_t& rhsOther, rhs_t& rhs
    ) {
        ...
    }
    ...
}
```

However, to directly employ the current CUDA implementation of parallel reduction, the Jacobians structure should be "lean enough" that a single thread can hold in its registers information about (at least) one full equation of (3). If this is not the case (e.g., with dense Jacobians), then the provided CUDA implementation of parallel reduction cannot be employed, and an alternative must be implemented.

# E    USE OF AI WRITING ASSISTANCE

We acknowledge the use of a large language model (LLM) to assist in refining the language, grammar, and clarity of this manuscript. All content was originally drafted by the human authors, and all AI-generated suggestions were critically reviewed, edited, and approved by the authors, who retain full responsibility for the final text.

