# OpenReview forum: "ParaRNN: Unlocking Parallel Training of Nonlinear RNNs for Large Language Models"
_ICLR.cc/2026/Conference — ICLR 2026 Oral_

### Official Review · Reviewer_M9m5 · 2025-10-31

**Soundness:** 4
**Presentation:** 4
**Contribution:** 3
**Rating:** 6
**Confidence:** 4

**Summary:**

This paper adds to the body of literature on efficient recurrent neural networks (RNNs). Although RNNs have potential for inference efficiency, particularly at long sequence lengths, the training of RNNs has been difficult due to the inherently sequential nature of the forward pass, unlike a Transformer forward pass which can be trivially parallelized.

This paper offers a new approach to parallelizing the training of RNNs, FlashRNN. The whole forward pass is solved in parallel via a fixed-point iterative method. This allows the efficient training of non-linear RNN cells, albeit the method is only tractable if the RNN cell has a Jacobian with a particular block-diagonal structure. The paper provides efficient implementation of the method in CUDA and scales FlashRNN to the 7B parameter scale.

**Strengths:**

+ The overall approach is a good contribution, and as far as I am aware this is the first work to allow training of non-linear RNNs via an exact parallelization method with no numerical instability.
+ The extensive implementation of a custom CUDA kernel which can be applied to a range of RNNs is a great contribution to the open-source community, and allows comparison with other architectures at a large scale.
+ The extensive computational results are quite impressive, including training a 7B parameter model from scratch in order to compare the performance of FlashRNN at language modelling tasks.
+ The paper is up-front and clear about the limitations of FlashRNN in section 2.1

**Weaknesses:**

+ The conceptual contributions of the paper could be argued to be somewhat limited. As far as I can tell (although I'm not an expert in this exact subject), the parallel solution of the sequence model by forward substitution is not novel and is described in Gonzales et al, Lim et al, etc. It is a bit hard to tell which of section 2 is novel and which is from previous work--we know that at least some of section 2 is previous work from the description, but it's not clear if it all is previous work. It seems like the conceptual contribution of this work could be characterized as following Gonzales et al, but simplifying the RNN cell instead of trying to solve the recurrence for general RNNs with a clever method.

+ The paper doesn't empirically justify the diagonalization of the RNN mixer in equation (9). While it's true that diagonalized matrices are often used in SSMs, the motivation for this work is that SSMs are not necessarily expressive enough. Therefore the justification for using a diagonalized matrix is not internally consistent. I think the authors could perhaps add extra ablations on the table 1 experiments where they use a non-diagonal GRU/LSTM (trained sequentially), to demonstrate that there is not much difference between diagonalized and non-diagonalized RNNs.

+ [minor] The paper is missing some earlier work on linear RNNs and parallelizations of linear RNNs, particularly [1,2,3]


  [1] Bradbury, James, et al. "Quasi-recurrent neural networks." arXiv preprint arXiv:1611.01576 (2016).

  [2] Martin, Eric, and Chris Cundy. "Parallelizing linear recurrent neural nets over sequence length." ICLR 2018

  [3] Qin, Zhen et al, Hierarchically Gated Recurrent Neural Network for Sequence Modeling, NeurIPS 2023

**Questions:**

+ Could you demonstrate that the diagonalization in equation (9) does not significantly affect the expressivity of the RNN layer?
+ Could you elaborate on why it is important to have a nonlinearity in the sequence mixer specifically, given the use of multiple layers with feature-mixing MLPs in all modern architectures? It is not clear to me, given the strong results of Mamba on most tasks (when equipped with MLPs, not just evaluated as a single layer).

---

> ### Author Response · Authors · 2025-11-21
>
> ## Weaknesses
>
> ---
> > __W1__ *The conceptual contributions of the paper could be argued to be somewhat limited [...]*
>
> We agree with the reviewer that our main contributions are not to the development of the method itself, but rather __pertain to the other strengths the reviewer has highlighted__, and particularly:
> - We are the first to validate this method on real-world LLM applications
> - We are the first to realize a viable way to scale this method to train models beyond the few-million parameters, pushing it all the way 7B
> - Perhaps most importantly, we are empowering practitioners to freely experiment on this method by open sourcing a robust, flexible, and efficient implementation
>
> Regarding the method itself, its inception dates back to 1960 [1] for application to parallel integration of ODEs, and its specific implementation and application to ML and RNNs had been proposed in [2,3]. Our main contribution in this sense lies in opting to directly simplify the structure of the RNN itself over using quasi-Newton methods as in [4], with the goal to boost stability when preparing for large-scale experiments. In Contribution 1 we make clear that we do not invent the method, but following the reviewer’s suggestion we will adapt the main text to further clarify how our work relates to the literature.
>
> - [1] Nievergelt et al, Parallel methods for integrating ordinary differential equations  http://dx.doi.org/10.1145/355588.365137
> - [2] Danieli et al., DeepPCR: Parallelizing Sequential Operations in Neural Networks https://arxiv.org/abs/2309.16318
> - [3] Lim et al., Parallelizing non-linear sequential models over the sequence length https://arxiv.org/abs/2309.12252
> - [4] Gonzalez et al, Towards Scalable and Stable Parallelization of Nonlinear RNNs https://arxiv.org/abs/2407.19115v1
>
> ---
> >__W2__ *The paper doesn't empirically justify the diagonalization of the RNN mixer in equation (9). While it's true that diagonalized matrices are often used in SSMs, the motivation for this work is that SSMs are not necessarily expressive enough. Therefore the justification for using a diagonalized matrix is not internally consistent. I think the authors could perhaps add extra ablations on the table 1 experiments where they use a non-diagonal GRU/LSTM (trained sequentially), to demonstrate that there is not much difference between diagonalized and non-diagonalized RNNs.*
>
> We would like to clarify that it is not currently possible to train dense RNNs at scale (at least with current methodologies). Please note that virtually every modern RNN relies on structure (specifically, block-diagonal) to make its large-scale training tractable. Ultimately, this is guided by necessity, and akin to employing multiple heads in attention.
>
> This being clarified, we fully agree that imposing structure can affect expressivity, but the *ideal* alternative of fully dense, unstructured RNNs is not yet a *real* one. __Our main point is that our work, while not yet allowing for the use of unstructured matrices, at least allows for the additional expressivity given by sequence-wise nonlinearities__. The usefulness of this is outlined in the synthetic tasks in Tab1,3, where diagonal nonlinear cells already achieve 100% accuracy, beating the diagonal linear Mamba.
>
> We also agree with the reviewer that expanding on the synthetic tasks in Tab1 and Tab6 would further clarify the gap between the feasible case of diagonal matrices and the ideal case of dense matrices. To this end, we additionally include the following tasks from the xLSTM paper and Merrill et al:
>
> |Model Type|Cycle Nav|Mod Arithmetic|A5|Copy Memory|
> |-|-|-|-|-|
> |GRU|100%|100%|100%|100%|
> |LSTM|100%|100%|100%|100%|
> |FlashGRU|97%|90%|40%|63%|
> |FlashLSTM|95%|94%|38%|67%|
> |Mamba2|57%|44%|36%|55%|
> |Transformer|100%|57%|28%|100%|
>
> The trend is overall maintained: nonlinear RNNs can solve these tasks much better than linear RNNs, although structuring the cell generally incurs a loss of performance.
>
> - [1] FlashRNN: I/O-Aware Optimization of Traditional RNNs on modern hardware, https://arxiv.org/abs/2412.07752
> - [2] Beck et al, xLSTM: extended long-short term memory, https://arxiv.org/abs/2405.04517
>
> ---
> >__[minor]__ *The paper is missing some earlier work on linear RNNs and parallelizations of linear RNNs, particularly [...]*
>
> We thank the reviewer for pointing out the relevant literature. We will also include the following relevant work:
> - Chaubard et al. Scaling Recurrent Neural Networks to a Billion Parameters with Zero-Order Optimization
> - Walker et al, Structured Linear CDEs: Maximally Expressive and Parallel-in-Time Sequence Mode
> - Gonzalez et al, Predictability Enables Parallelization of Nonlinear State Space Models

---

> ### Author Response · Authors · 2025-11-21
>
> ## Questions
>
> > __Q1__ *Could you demonstrate that the diagonalization in equation (9) does not significantly affect the expressivity of the RNN layer?*
>
> We refer to our answer to Weakness 2, and also reply to Weakness 1 of reviewer BeBu
>
>
> ---
> > __Q2__ *Could you elaborate on why it is important to have a nonlinearity in the sequence mixer specifically, given the use of multiple layers with feature-mixing MLPs in all modern architectures? It is not clear to me, given the strong results of Mamba on most tasks (when equipped with MLPs, not just evaluated as a single layer).*
>
> Hidden states of linear SSMs like Mamba can only evolve by exponential decay (see [1], but the intuition is rooted in the type of trajectories that can be followed by linear ODEs— which Mamba represents). This has a direct repercussion on the expressivity of the cell itself (as illustrated also in the synthetic tasks in Tab1,6). Indeed, much of the expressivity of Mamba can be attributed to its “external” components (see [3] about the role of input selectivity, gate and pre-convolution). By contrast, including nonlinearities allows for much more flexibility in the behaviors the cell can track (see also [2]).
>
> This being said, we fully agree that the extent to which this extra flexibility is relevant to the model performance will depend on the task at hand, and the question of how to best leverage this extra flexibility is very much an open one. Nonetheless, with our paper we aim to demonstrate that such flexibility (previously only limited to small-scale models) is still accessible also at scale, and to provide the tools (in the form of a scalable, hardware-efficient codebase) to progress research in this direction.
>
> - [1] Cirone et al., Theoretical foundations of deep selective state-space models
> - [2] Merrill et al, The illusion of state in state-space models
> - [3] Huang et al, Understanding input selectivity in Mamba: Impact on approximation power, memorization, and associative recall capacity

---

> > ### Comment · Reviewer_M9m5 · 2025-11-24
> >
> > Thanks for your replies, and experimental results, they have clarified what trade-offs the FlashRNN diagonalization requirement is imposing.
> >
> > The intuition about SSM cells having linearly decaying hidden state is very useful to understand the contribution of this work.
> > I'll increase my recommendation.

---

### Official Review · Reviewer_bSyi · 2025-11-03

**Soundness:** 3
**Presentation:** 3
**Contribution:** 3
**Rating:** 8
**Confidence:** 4

**Summary:**

The paper proposes FlashRNN, which parallelizes nonlinear RNN application by recasting the full sequence recurrence as a single nonlinear system (Eq. (2)) and solving it with Newton iterations whose linear subproblems have a block-bidiagonal structure solvable via parallel scan.

**Strengths:**

1. Novelty:  Prior work typically applies Newton/scan to given RNNs (e.g., Lim; Gonzalez). This paper instead redesigns LSTM/GRU so their Jacobians are diagonal or 2×2 block-diagonal, which makes each Newton step’s linear system amenable to efficient parallel reduction without runtime Jacobian approximations.
2. Scale of Experiments: This is the most exciting part. The paper successfully trains 7B-parameter FlashGRU/FlashLSTM models and reports their downstream task results. This proves that FlashRNN is not just a toy model. Moreover, the efficient CUDA implementation further highlights the significance of this contribution.

**Weaknesses:**

N/A

**Questions:**

N/A

---

> ### Author Response · Authors · 2025-11-23
>
> We thank the reviewer for their positive impression on our work, and fully share their enthusiasm regarding our key contributions. Indeed we believe that demonstrating competitive performance at the 7B scale is crucial for establishing  (again!) RNNs as a practical alternative for the community, and confirm that the parallelization method employed is more than just a theoretical contribution. We are also grateful for the appreciation of the engineering effort that went into the development of the FlashRNN codebase, which we invested to guarantee both deployment-level efficiency and ease of adoption by practitioners.
>
> The score provided by the reviewer is already extremely encouraging, but we remain available to implement any recommendation which might further improve the quality of our work, either by clarifying its presentation, or by considering additional analyses. To this end, we flag that (following other reviewers’ suggestions) we are already adding to the paper:
> - A detailed profiling of the memory consumption required by the FlashRNN cells, confirming linear scaling in both hidden size and sequence length, and per-sample memory usage comparable to other architectures for the models considered
> - An analysis of the capabilities of our cells on more synthetic tasks, including comparisons against dense architectures to better quantify the impact of diagonalizing the Jacobian structure
> - An expansion of the results on token throughput, to confirm that similar behavior hold across scales
>
> We remain open to additional suggestions, and thank the reviewer again for their time and their encouraging words.

---

### Official Review · Reviewer_5Qbd · 2025-11-03

**Soundness:** 3
**Presentation:** 2
**Contribution:** 3
**Rating:** 6
**Confidence:** 3

**Summary:**

The paper introduces FlashRNN, a framework that parallelizes training-time application of nonlinear RNNs by recasting the unrolled recurrence as a single all-at-once system solved with Newton iterations and a custom parallel reduction (prefix-scan) solver over a block bi-diagonal linearization. Implementations span (i) pure PyTorch, (ii) CUDA-accelerated reduction, and (iii) a fully fused CUDA kernel.

The authors adapt GRU and LSTM (FlashGRU/FlashLSTM) to make their Jacobians diagonal or 2×2 block-diagonal, enabling efficient reductions. They report large speedups over naïve sequential application versus traditional RNNs and show competitive perplexity and downstream results vs. Mamba and a Transformer baseline at 7B parameters.

**Strengths:**

The proposed idea casting nonlinear RNN application into a Newton+scan routine with a specialized CUDA solver is well motivated and clearly explained.

Thoughtful GPU hierarchy design, e.g. Appendix D2.

The author promised code release which would encourage the community try the proposed method.

**Weaknesses:**

It's unclear to me how the the diagonal (GRU) and block-diagonal (LSTM) Jacobians limitation been overcomes. Any study shows those limitation not matters at scale?

I'd suggest add more opensource baseline in Table 2.

It's unclear numerical stability when further scale up the model.

It would be interesting to see more ablation on newton iterations, how does the memory footprint change as context length change, and how far the context length can be pushed.

**Questions:**

See the weaknesses section.

---

> ### Author Response · Authors · 2025-11-21
>
> >__W1__ *It's unclear to me how the diagonal (GRU) and block-diagonal (LSTM) Jacobians limitation been overcomes. Any study shows those limitation not matters at scale?*
>
> We point out that __having to use (block-)diagonalization of state matrices is not a specific limitation of our work__, but one that has been plaguing the development of RNNs in general. As an indication of this, please note that (to our knowledge) all modern RNNs also limit the amount of mixing happening within the hidden state components, and rely on the very same (block-)diagonal structure we leverage. This is because a high degree of component mixing increases not only storage requirements, but also the overall cost of applying the RNN cell. Despite the impact on expressivity given by the diagonalization of the RNN cell (which we exemplify with synthetic experiments in Tab1 and Tab6, see also reply to W1 of BeBu), __imposing structure is necessary for large scale training__. Empirical work exploring the capabilities of dense RNNs at scale is limited precisely because of this: we simply cannot train them. For theoretical work in this sense, we point out  Merrill et al., which flags both nonlinearity and density as useful in boosting an RNN's expressivity. Our work, we iterate, focuses on __allowing for nonlinearities to be employed__, which wasn't considered feasible up until now.
>
>
> ---
> >__W2__ *I'd suggest add more opensource baseline in Table 2*
>
> Additional baselines --while indeed helpful to clarify the performance of our models-- come at a significant computational cost (especially at 7B regimes). Since the focus of our paper is to showcase the feasibility of training RNNs at scale to achieve competitive performance in language modelling, and not setting state-of-the-art on any particular benchmark, we believe that the existing comparisons are sufficient to substantiate the claims made in the paper, ie, that we can achieve competitive performance with existing LLMs.
>
>
> ---
> >__W3__ *It's unclear numerical stability when further scale up the model*
>
> In our paper we are already reporting results for __scales ranging from 400M to 7B__ (see Fig1 and Tab5). These show consistent scaling, and we haven't experienced numerical instabilities in training our models. We believe these results provide sufficient evidence for the stability of our training procedure at scale. For context, 7B is already beyond the scales showcased in the Mamba, Mamba2, and xLSTM papers (capping at ~2.8B), and on par with the scales showcased in the newest xLSTM-7B paper.
>
> ---
> >__W4__ *It would be interesting to see more ablation on newton iterations, how does the memory footprint change as context length change, and how far the context length can be pushed*
>
> - *Newton stability and context length* Please refer to AppA in the main paper. Particularly, in Fig4 __we report results on Newton convergence when varying model type__ (FlashGRU/FlashLSTM), __stage of training__ (initialization/end of training), __and sequence length__ (up to 2^{11}). Overall, there is __no visible variation in the convergence behavior__, which seems to hold stably regardless of context length. The only noticeable difference is between the training stages: towards end of training, convergence generally requires one more iteration to reach machine precision than at initialization. If the reviewer has specific requests about additional ablations, we would be happy to include them.
>
> - *Memory footprint* The parallel application of the cell requires memory __growing linearly with sequence length__. Notice that this is __by design__: any RNN requiring worse-than-linear memory scaling would forfeit one of its main advantages over Transformers, which is why we were conscious of memory consumption in our implementation. To further confirm that this scaling holds in practice, here we report a table with peak memory utilization in [MB] when parallel-applying a cell of hidden state size $d_H=2^{10}$ to an input of varying sequence length:
>
> | Cell Type \ $L$ | $2^1$  | $2^2$   | $2^3$  | $2^4$  | $2^5$  | $2^6$  | $2^7$  | $2^8$  | $2^9$   | $2^{10}$ |
> |-----------------|--------|---------|--------|--------|--------|--------|--------|--------|---------|----------|
> | FlashGRU        | 0.0586 | 0.1055  | 0.1992 | 0.3867 | 0.7617 | 1.5117 | 3.0117 | 6.0195 | 12.0117 | 24.0117  |
> | FlashLSTM       | 0.0664 | 0.1211  | 0.2305 | 0.4492 | 0.8867 | 1.7617 | 3.5117 | 7.0117 | 14.0117 | 28.0117  |
>
> showing a clear linear scaling.

---

### Official Review · Reviewer_BeBu · 2025-11-04

**Soundness:** 3
**Presentation:** 4
**Contribution:** 4
**Rating:** 6
**Confidence:** 3

**Summary:**

This paper introduces *FlashRNN*, a framework that reformulates the entire recurrent computation in nonlinear RNNs over a sequence as a *single nonlinear system of equations*, which can then be solved in parallel via Newton iterations and prefix-scan parallel reductions.  To make this practical, the authors constrain the RNN Jacobians' structure (e.g., diagonal or block-diagonal) so that the prefix-scan remains efficient and take care of the channel mixing via a MLP, similarly as done in SSMs. They further provide a CUDA implementation that automatically parallelizes user-defined RNN cells and demonstrate training of 7B-parameter nonlinear RNNs (FlashGRU, FlashLSTM) with competitive perplexity to Transformers and Mamba2 while achieving significant (up to 665×) speedups over naïve sequential RNNs.

**Strengths:**

1. Tackling the core sequential bottleneck of nonlinear RNNs is a long-standing challenge, so the motivation of the paper is clear. The provided solution, i.e. recasting recurrence evaluation as a global nonlinear system of equations and applying Newton iterations combined with parallel prefix reductions, is conceptually clean and theoretically grounded from previous work.

2. The authors release **FlashRNN**, a PyTorch + CUDA framework that generalizes to arbitrary RNN cells, lowering adoption barriers and fostering community experimentation.

3. The authors train the efficient versions of GRUs and LSTMs up to 7B parameters with substantial wall-clock speedups. Few works have shown nonlinear RNNs trained at this scale and doing so has potential to reshape how sequence models are trained.

4. The experimental (including runtime) analysis is thorough. The authors provide profiling, ablations, and comparisons to Transformers and SSMs, which add credibility.

5. The authors openly discusses convergence challenges, overheads, and structural constraints in the Jacobian.

**Weaknesses:**

1. As the authors write, imposing diagonal or block-diagonal Jacobians simplifies parallelization but may severely limit expressivity due to channel mixing. The solution taken is similar as in SSMs, i.e. using downstream MLPs to "restore" expressivity. I believe this needs stronger empirical support on more state-tracking tasks where xLSTM performs relatively well.

2. The authors assume that a small, fixed number of Newton iterations (e.g., 3) suffice. However, this is not guaranteed across other recurrent architectures not studied in this paper or different tasks (e.g., time series forecasting, speech, RL, etc.). Where there any scenarios where convergence fails or oscillates? Otherwise, formal guarantees on truncated Newton convergence or stability under limited precision would benefit the paper.

3. Newton iterations can be sensitive to numerical precision.

4. While the authors briefly claim reduced memory scaling relative to BPTT, the paper lacks a quantitative analysis of GPU memory usage across model sizes.

**Questions:**

1. How difficult is it for practitioners to define a new RNN cell compatible with FlashRNN’s structural constraints?

2. Have you tried experiments with FlashRNN in mixed or low precision?

3. Do you plan and how easy would it be to integrate fused CUDA kernels from [Pöppel et al. (2024)](https://arxiv.org/pdf/2412.07752) in FlashRNN?

4. Would the quasi-Newton approximations as in [Gonzalez et al. (2024)](https://arxiv.org/pdf/2407.19115) work also at scale instead of imposing the diagonal matrix structure?

5. How do GPU memory and throughput scale with hidden size?

---

> ### Author Response · Authors · 2025-11-21
>
> ## Weaknesses
>
> ---
> >__W1__ *As the authors write, imposing diagonal or block-diagonal Jacobians simplifies parallelization but may severely limit expressivity due to channel mixing. The solution taken is similar as in SSMs, i.e. using downstream MLPs to "restore" expressivity. I believe this needs stronger empirical support on more state-tracking tasks where xLSTM performs relatively well.*
>
> The reviewer raises an excellent point. To address this, we expand Tab1,6 with additional preliminary results on more synthetic tasks taken from the xLSTM paper and Merrill et al, and include a direct comparison with dense RNNs:
>
> |Model Type|Cycle Nav|Mod Arithmetic|A5|Copy Memory|
> |-|-|-|-|-|
> |GRU|100%|100%|100%|100%|
> |LSTM|100%|100%|100%|100%|
> |FlashGRU|97%|90%|40%|63%|
> |FlashLSTM|95%|94%|38%|67%|
> |Mamba2|57%|44%|36%|55%|
> |Transformer|100%|57%|28%|100%|
>
> In general, diagonalization does affect the overall accuracy. Particularly for A5 (a representative of NC1) a dense cell allows for perfect solution. To better contextualize these results, however, we want to highlight the __dramatic increase in performance of nonlinear RNNs over linear RNNs__ such as Mamba. This is the main motivation behind our paper: including sequence-wise nonlinearities boosts RNN expressivity. In particular, it is not our goal to demonstrate that the specific diagonal structure we employ is the best one possible: this is just one possible simplification to allow computational tractability, and one shared by virtually all modern RNNs.
>
> ---
> >__W2__ *The authors assume that a small, fixed number of Newton iterations (e.g., 3) suffice. However, this is not guaranteed across other recurrent architectures not studied in this paper or different tasks (e.g., time series forecasting, speech, RL, etc.). Where there any scenarios where convergence fails or oscillates? Otherwise, formal guarantees on truncated Newton convergence or stability under limited precision would benefit the paper.*
>
> We fully share the reviewer’s concerns on the generalizability of Newton’s application, which is why in Sec2.1 we flagged this as a limitation. More than the specific task at hand, Newton stability is mainly linked to the behaviour of the RNN cell Jacobians: if these are allowed to vary wildly and explode, then one cannot expect good convergence properties from Newton: some formal guarantees and theoretical results in this sense are available in recent work [1]. In practice, then, one should ensure that the Jacobians spectral norm is bounded in some sense. Nonetheless, we point out that the __requirements for ensuring fast Newton convergence do overlap with those to ensure training stability__ in modern RNNs: Jacobians should be controlled anyway, because unbounded Jacobians can result in exploding gradients. Mamba itself hard-codes this in the parameterization of its state matrix, which effectively bounds its eigenvalues in (0,1].
>
> [1] Gonzalez et al: Predictability Enables Parallelization of Nonlinear State Space Models, https://arxiv.org/abs/2508.16817
>
> ---
> >__W3__ *Newton iterations can be sensitive to numerical precision*
>
> The main issue lies in the system stability: if small perturbations (due to numerical approximations, or noise) are enough to push the system outside the convergence basin of Newton, then the system is unsuited to be solved via Newton to begin with (because one can’t prevent numerical approximations from piling up). If instead the reviewer is concerned about numerical approximations affecting the solution even in the stable case, we point out that also applying the RNN sequentially necessarily introduces approximation errors (scaling with sequence length).
>
> ---
> >__W4__ *While the authors briefly claim reduced memory scaling relative to BPTT, the paper lacks a quantitative analysis of GPU memory usage across model sizes*
>
> We __do not claim reduced memory scaling relative to BPTT__: this claim would be necessarily false—the reason being, FlashRNN requires computing temporary quantities for all tokens at once, rather than one at a time, thus necessarily incurring in larger peak memory consumption than in the sequential case. This is a common trade-off in parallelization. We invite the reviewer to point out the parts of the manuscript causing the confusion, as they should be corrected to avoid misunderstandings.
>
> Following the reviewer’s recommendation, in Tab3 we have __included an additional column stating the per-sample peak memory usage at training__ for each model considered in the paper. The measurements report [GiB/sample]:
>
> |Model Type\Size|400M|1B|3B|7B|
> |-|-|-|-|-|
> |FlashGRU|6.2|9.8|16.8|38.2|
> |FlashLSTM|6.2|9.8|16.9|39.3|
> |Mamba2|5.2|9.1|17.2|39.8|
> |Transformer|5.3|8.5|14.1|31.0|
>
> Our results confirm that __memory consumption is balanced among all models types and sizes__, with Mamba2 being slightly more memory-intensive, and Transformer slightly less (see also reply to W4 of reviewer 5Qbd).

---

> ### Author Response · Authors · 2025-11-21
>
> ## Questions
>
> ---
> >__Q1__ *How difficult is it for practitioners to define a new RNN cell compatible with FlashRNN’s structural constraints?*
>
> Not difficult at all, as indeed ease-of-use was one of our driving goals in the codebase design. For reference, here’s the main function for implementing the FlashLSTM cell used in our paper (simplified for clarity):
> ```
> class FlashLSTM(RNNCellBlockDiagImpl[FlashLSTMSystemParameters]):
>   @classmethod
>   def recurrence_step(
>       cls,
>       x: torch.Tensor,
>       h: torch.Tensor,
>       system_parameters: FlashLSTMSystemParameters
>   ) -> torch.Tensor:
>     cc, hh = cls._split_hidden_state(h)        # split the collated [c,h]
>     f = system_parameters.nonlin_f(
>       Af * hh + Cf * cc + torch.einsum('ij,...j->...i',(Bf, x)) + bf
>     )
>     c = system_parameters.nonlin_c(
>       Ac * hh + torch.einsum('ij,...j->...i',(Bc, x)) + bc
>     )
>     cc = f * cc + (1 - f) * c
>     o = system_parameters.nonlin_o(
>       Ao * hh + Co * cc + torch.einsum('ij,...j->...i',(Bo, x)) + bo
>     )
>     hh = o * system_parameters.nonlin_h(cc)
>     return cls._recombine_hidden_state(cc, hh)  # collate [c,h]
> ```
> To impose the desired structure, it suffices to:
> - Inherit from the correct RNNCellBlockDiagImpl specialization
> - Ensure that the recurrence step satisfies the block-diagonal structure ($A_*,C_*$ above are diagonal matrices stored as vectors, so no mixing of components occurs)
>
> ---
> >__Q2__ *Have you tried experiments with FlashRNN in mixed or low precision?*
>
> Yes, and the code we provide does support also bfloat16 and float16 datatypes. In practice, we use automated mixed precision (ie upcast to float32) for the parallel reduction operations, in line with best practices for NN training.
>
> ---
> >__Q3__ *Do you plan and how easy would it be to integrate fused CUDA kernels from Pöppel et al. (2024) in FlashRNN?*
>
> Integration is feasible: suffices to provide another specialization among the `RNNCellApplication` classes in our code, responsible of directly invoking Pöppel’s kernels. The main challenge would probably be syncing the structures holding the RNN parameters (`system_parameters`, in our code).
>
> Incidentally, to avoid misappropriation and confusion with Pöppel’s work, we will change our project’s name from FlashRNN to ParaRNN.
>
> ---
> >__Q4__ *Would the quasi-Newton approximations as in Gonzalez et al. (2024) work also at scale instead of imposing the diagonal matrix structure?*
>
> We didn’t investigate this extensively, as we were looking for a more robust approach. Gonzalez et al. (2024) themselves point out that using quasi-Newton with diagonal approximations requires additional stabilization (notice the reported iterations >1k in Tab2 and Fig4 in their paper). Nonetheless, applying a quasi-Newton method instead of simplifying the RNN cell remains a viable option, particularly if guarantees are provided that justify the approximation (eg, diagonal dominance).
>
> ---
> >__Q5__ *How do GPU memory and throughput scale with hidden size?*
>
> __Memory__ Peak GPU memory utilization scales linearly with both sequence length and hidden size. This is because the application of the algorithm requires the assembly of the whole linearized system. Of course this only holds if RNN cells with structured Jacobians are considered: for fully unstructured Jacobians, the memory requirements will scale quadratically in hidden size. To confirm the linear scaling, we report a table with peak memory utilization [MB] when parallel-applying a cell of varying hidden state sizes to a sequence of length $L=2^8$:
>
> |Cell Type \ $d_h$|$2^2$|$2^3$|$2^4$ |$2^5$|$2^6$|$2^7$|$2^8$| $2^9$|$2^{10}$|$2^{11}$|$2^{12}$|$2^{13}$|$2^{14}$|$2^{15}$|
> |-|-|-|-|-|-|-|-|-|-|-|-|-|-|-|
> |FlashGRU|0.094|0.188|0.375|0.750|1.50|3.00|6.00|12.00|24.01|48.02|96.05|192.10|384.19|768.38|
> |FlashLSTM|0.110|0.219|0.438|0.875|1.75|3.50|7.00|14.01|28.01|56.02|112.05|224.09|448.19|896.38|
>
> __Throughput__ The diagonal structure of the cells we consider make their application perfectly parallel across hidden state components: consequently, their application runtime is theoretically not affected by hidden state dimension. In practice, we need to take into account that (i) The MLPs connecting the RNN cells must also scale with the cell hidden state dimension, and their application time does scale with dimensionality; (ii) GPU capacity is limited, and component-wise operations will get sequentialized after a certain dimension. To address the reviewer’s concern, we provide throughput results at all scales: at regime ($L=2^{11}$), we have
> |Model Type \ Size|400M|1B|3B|7B|
> |-|-|-|-|-|
> |FlashGRU|45.59|42.37|34.50|33.52|
> |FlashLSTM|40.43|40.01|31.94|30.90|
> |Mamba2|34.35|34.40|25.43|22.35|
> |Transformer|14.01|4.77|2.47|1.07|
> The biggest decrease in throughput occurs when jumping from 1B to 3B, when we double the number of layers in the model (see Tab3). Doubling hidden state size instead has only marginal impact on throughput (mainly attributed to the MLPs).

---

> > ### Comment · Reviewer_BeBu · 2025-11-22
> >
> > Thank you for your detailed response. I have adjusted my score accordingly since you have addressed my concerns. I am convinced that this work is relevant to the community, especially since it shows scalability of traditional models, such as LSTM and GRU, to both large context length and model size, with performance on par with Transformers at the same scale. Furthermore, the framework is agnostic to the RNN type, so it can be broadly adapted to new RNN architectures.
> >
> > I recommend the authors, as they also suggested, to change the title in order to avoid confusion to the FlashRNN work from Pöppel et al. Finally, I hope that you will incorporate the new experiments in the paper and add a discussion on state-tracking capabilities and expressivity in general, wherer you also discuss recent findings [1, 2] that linear RNNs with negative eigenvalues in their state-transition matrix are expressive enough for these tasks.
> >
> > [1] Grazzi et al. Unlocking state-tracking in linear rnns through negative eigenvalues. In ICLR 2025 (Oral).
> >
> > [2] Siems et al. Deltaproduct: Improving state-tracking in linear rnns via householder products. In NeurIPS 2025.

---

### Author Response · Authors · 2025-12-01
**Rebuttal summary**

Dear Area Chair,

We are reaching out in light of the recent changes to the ICLR review process. We understand the confidentiality breach forced the Program Chairs into a difficult situation, but we are concerned their decisions might disproportionally affect the evaluation of our paper.

Though our work already received positive initial reviews (6,6,6,8), the rebuttal phase was exceptionally productive, leading two reviewers to raise their scores (6→8), which positioned our paper very strongly ([56th overall](https://papercopilot.com/statistics/iclr-statistics/iclr-2026-statistics/)). We are then concerned that reverting to pre-rebuttal scores might overlook these critical improvements.

To this end, we want to summarize the main concerns raised by the reviewers, and how we addressed them:

* **Expressivity of (block-)diagonal RNNs** As a simplification to ensure computational tractability of our method, we considered (block-)diagonal formulation of common RNN cells. The reviewers were concerned this might affect the overall expressivity of the RNNs. To better quantify this impact, we **included baselines comparing against dense nonlinear RNNs on additional synthetic tasks**. Moreover, we clarified that our approach is not at all tied to a diagonal simplification: the main focus remains the superior expressivity of _nonlinear_ RNN cells over _linear_ ones — we only considered diagonalization as it represent the most common structure in modern RNN architectures design, including Mamba and xLSTM.
* **Memory Utilization** Given the necessity for the solvers employed in our work to materialize intermediate variables (namely, cell Jacobians and residuals), the reviewers were concerned about the overall memory budget required by FlashRNN. We clarified that *by design* this scales linearly with sequence length and hidden state size, and further confirmed this scaling by **including explicit memory profiling for the cells application**. In particular, results showed per-sample memory consumption comparable to Transformers and Mamba.
* **Newton Stability** The effectiveness of FlashRNN is tied to the (fast) convergence of Newton iterations, which we employ to compute the nonlinear cells application in parallel. Due to this, the reviewers were concerned about the generalizability of our approach. While this is indeed a potential limitation (which we ourselves highlight in Sec2.1), we **included pointers to relevant recent work which ties fast Newton convergence to the boundedness of the cell Jacobians**. Particularly, this implies that the very same stabilization techniques used to prevent gradient explosion in vanilla RNNs sequential training can simultaneously benefit their parallelizability. On top of this, AppA already reports extensive ablation studies on Newton convergence, showcasing its robustness to varying sequence lengths, stage of training, and cell considered.

It is also important to highlight the significant praise that our contributions received from the reviewers, who appreciated in particular:

* **The impact of showcasing for the first time parallel trainability for large-scale nonlinear RNNs** (BeBu: “potential to reshape how sequence models are trained”, bSyi: “successfully trains 7B [...] models [...]. This proves that FlashRNN is not just a toy model”, M9m5: “first work to allow training of non-linear RNNs via an exact parallelization method”, “computational results are quite impressive, including training a 7B parameter model” )
* **The quality of our codebase and its potential for accelerating future research** (BeBu: “authors release FlashRNN [...] lowering adoption barriers, 5Qbd: ”Thoughtful GPU hierarchy design“, ”code release [...] encourage[s] the community to try the proposed method“, bSyi ”efficient CUDA implementation further highlights the significance of this contribution“, M9m5 ”a great contribution to the open-source community“)
* **The transparency with respect to the limitations of the method** (BeBu: ”openly discusses convergence challenges, overheads, and structural constraints“, M9m5: ”up-front and clear about the limitations“)

---

> ### Author Response · Authors · 2025-12-01
>
> The impact of the responses we provided during the rebuttal phase was clearly reflected in the reviewers’ final evaluations, bringing two of them to raise their scores. More in detail:
>
> * **Reviewer BeBu raised score from 6 to 8**, due to the inclusion of memory scaling results, and of the additional synthetic tasks on state-tracking capabilities
> * **Reviewer M9m5 also raised score from 6 to 8**, acknowledging the clarification on the impact of nonlinearities on the cell expressivity, and the inclusion of additional experimental results
> * **Reviewer bSyi maintained their strong score of 8** having praised from the beginning the practical contributions of the paper in achieving parallel trainability of nonlinear RNNs up to 7B parameters
> * **Reviewer 5Qbd maintained a score of 6**. We note this reviewer had minimal engagement in the reviewing process, and their concerns about stability at scale (addressed by our experiments at 7B scales) and about Newton stability (addressed by the extensive ablation study in AppA) were also covered in the rebuttal.
>
> In light of all this, we respectfully request that, as you deliberate on our paper,  you take into account all the improvements that were brought to the paper thanks to the rebuttal. We strongly believe on the significance of our work, and hope it will not be negatively impacted by these unforeseen circumstances.
>
>
> Thank you for your time and understanding,
>
> The Authors

---

### Meta-Review · Area_Chair_auTL · 2025-12-20

**Summary:**

This paper presents FlashRNN, a framework for enabling the parallel training of large-scale, nonlinear Recurrent Neural Networks (RNNs). The work addresses a long-standing bottleneck in sequence modeling by recasting the sequential recurrence as a single system of equations solvable with Newton's method and parallel scan operations. The authors demonstrate the viability of this approach by successfully training 7-billion-parameter GRU and LSTM models that achieve performance competitive with modern Transformer and State Space Model (SSM) architectures.

The review process was highly dynamic and ultimately very positive. Initial scores were [6, 6, 6, 8]. Following an exceptionally productive rebuttal period where the authors provided substantial new experiments and clarifications, two reviewers raised their scores, resulting in a final configuration of **[6, 8, 8, 8]**. The consensus among the engaged reviewers is a strong accept.

This paper makes a significant practical contribution by demonstrating the feasibility of training nonlinear RNNs at a scale previously thought impractical. The open-source, high-performance CUDA implementation is a valuable asset for the community, reopening a promising research direction. While the work has acknowledged limitations in conceptual novelty and theoretical guarantees, its empirical achievements and the quality of the engineering are substantial.

**Reviewer Concerns:**

The review process centered on four primary thematic concerns, which collectively informed the final decision. The authors' ability to comprehensively address most of these points was the key driver of the positive outcome.

| Concern Category | Key Issues Raised by Reviewers | Summary of Author's Rebuttal & Resolution |
| :--- | :--- | :--- |
| **Expressivity vs. Structure** | The constraint of using diagonal or block-diagonal Jacobians to ensure computational tractability might severely limit the model's expressive power, a concern raised by Reviewers BeBu, 5Qbd, and M9m5. This seemed internally inconsistent with the paper's motivation of overcoming the limitations of linear SSMs. | The authors provided new synthetic experiments comparing their diagonal models to dense RNNs, quantifying the performance trade-off. They successfully argued that this structural constraint is a pragmatic necessity common to virtually all modern large-scale RNNs (including Mamba and xLSTM) and that their primary contribution is enabling *nonlinearity*, which still offers a significant expressive advantage over linear models. This argument was accepted by the engaged reviewers. |
| **Convergence & Stability** | Reviewers BeBu and 5Qbd questioned the stability and generalizability of using a fixed, small number of Newton iterations. They sought stronger guarantees that the method would converge for other architectures, tasks, and at even larger scales, and raised concerns about numerical precision. | The authors acknowledged this as an inherent limitation of the method. They provided theoretical context linking convergence to bounded Jacobians (a common requirement for stable RNN training) and pointed to extensive empirical ablations in the appendix. The successful training of 7B models served as strong practical evidence of stability, which satisfied the reviewers. |
| **Memory & Scalability** | The original submission lacked a quantitative analysis of GPU memory usage. Reviewers BeBu and 5Qbd requested detailed profiling of how memory consumption scales with model size and context length. | The authors provided comprehensive memory profiling tables in the rebuttal. They clarified that while peak memory usage is necessarily higher than for sequential BPTT, it scales linearly with sequence length and hidden size. Crucially, they showed that the per-sample memory footprint is comparable to that of Transformer and Mamba2 models at all tested scales (400M to 7B), fully addressing this concern. |
| **Conceptual Novelty** | Reviewer M9m5 correctly pointed out that the core parallel solving method is not new, having been explored in prior work. The concern was whether the paper's conceptual contribution was limited to merely adapting existing methods. | The authors candidly agreed, positioning their contribution not as the invention of a new method, but as its first successful application and scaling to billion-parameter language models. They highlighted their key contributions as the empirical validation, the significant engineering effort to create a robust open-source framework, and the strategic choice to simplify the RNN structure for stability, which proved successful. This framing was accepted by the reviewer. |

**Reviewer Scores:**

The rebuttal was highly effective and directly led to two reviewers increasing their scores from 6 to 8.

**Concerns Fully Addressed by the Rebuttal:**
*   **Expressivity Justification (M9m5, BeBu):** The new synthetic experiments and theoretical clarification on why nonlinearities matter (i.e., escaping the linear ODE limitations of SSMs) were decisive. Reviewer M9m5 explicitly cited this as the reason for their score increase.
*   **Memory Consumption Analysis (BeBu, 5Qbd):** The addition of detailed memory profiling tables completely resolved all questions about memory footprint and scalability.
*   **Practical Implementation Questions (BeBu):** The authors provided clear code examples and explanations regarding the ease of use and support for mixed precision.

---

### Decision · Program_Chairs · 2026-01-26

Accept (Oral)